# Mathematical modeling links Wnt signaling to emergent patterns of metabolism in colon cancer

Mary Lee[1,†], George T Chen[2,†], Eric Puttock[1], Kehui Wang[3,4], Robert A Edwards[3,4], Marian L Waterman[2,4,5,*] & John Lowengrub[1,4,5,6,**] (ID)

## Abstract

Cell-intrinsic metabolic reprogramming is a hallmark of cancer that provides anabolic support to cell proliferation. How reprogramming influences tumor heterogeneity or drug sensitivities is not well understood. Here, we report a self-organizing spatial pattern of glycolysis in xenograft colon tumors where pyruvate dehydrogenase kinase (PDK1), a negative regulator of oxidative phosphorylation, is highly active in clusters of cells arranged in a spotted array. To understand this pattern, we developed a reaction–diffusion model that incorporates Wnt signaling, a pathway known to upregulate PDK1 and Warburg metabolism. Partial interference with Wnt alters the size and intensity of the spotted pattern in tumors and in the model. The model predicts that Wnt inhibition should trigger an increase in proteins that enhance the range of Wnt ligand diffusion. Not only was this prediction validated in xenograft tumors but similar patterns also emerge in radiochemotherapy-treated colorectal cancer. The model also predicts that inhibitors that target glycolysis or Wnt signaling in combination should synergize and be more effective than each treatment individually. We validated this prediction in 3D colon tumor spheroids.

**Keywords** glycolysis; spatial pattern; tumor metabolism; Warburg effect; Wnt signaling

**Subject Categories** Cancer; Quantitative Biology & Dynamical Systems; Signal Transduction

**Mol Syst Biol. (2017) 13: 912**

See also: **Z Dai & JW Locasale** (February 2017)

## Introduction

A hallmark feature of many cancers is "aerobic glycolysis", or the Warburg effect, a form of metabolism whereby cells skew their balance of cellular metabolism away from oxidative phosphorylation (OXPHOS) to favor glycolysis, despite the availability of sufficient levels of oxygen (Warburg, 1956). Cellular emphasis on Warburg metabolism is intriguing since it is much less efficient than OXPHOS in producing energy (four molecules of ATP produced by glycolysis for each molecule of glucose consumed versus 36 molecules by OXPHOS). Warburg metabolism has been hypothesized to be beneficial because glycolytic intermediates can be used as biosynthetic building blocks for cell growth and proliferation, suggesting that this mode of glucose utilization is essential for actively expanding tumors (Vander Heiden et al, 2009; Pavlova & Thompson, 2016). There are other effects as well: The production of lactate acidifies the tumor microenvironment, an environmental condition that can enhance tumor invasiveness (Gatenby & Gillies, 2004), and induces angiogenic responses for increased delivery of glucose, oxygen, and other nutrients (Végran et al, 2011), effects that are growth promoting and provide cancer cells with a fitness advantage.

Oncogenic, overactive Wnt signaling has been recently linked to metabolic and nutrient programming in tumors. For example, in colon cancer, Wnt signaling is proposed to increase expression of key glycolytic factors that enhance Warburg metabolism and angiogenesis (Pate et al, 2014). Oncogenic Wnt signaling most commonly derives from genetic inactivation of one or more signaling components (e.g., adenomatous polyposis coli, APC), inactivating mutations that cause the pathway to become chronically activated and to trigger overexpression of Wnt target genes. One such target gene is pyruvate dehydrogenase kinase 1 (PDK1), a mitochondrial kinase that inhibits the pyruvate dehydrogenase complex (PDC) via phosphorylation of the component pyruvate dehydrogenase (PDH) (Pate et al, 2014). Since PDC converts pyruvate to acetyl CoA for mitochondrial respiration, phosphorylation/inhibition of PDH by PDK1 suppresses OXPHOS modes of metabolism (ATP and $CO_2$

1 Department of Mathematics, University of California, Irvine, Irvine, CA, USA
2 Department of Microbiology and Molecular Genetics, University of California, Irvine, Irvine, CA, USA
3 Department of Pathology, School of Medicine, University of California, Irvine, Irvine, CA, USA
4 Chao Family Comprehensive Cancer Center, University of California, Irvine, Irvine, CA, USA
5 Center for Complex Biological Systems, University of California, Irvine,Irvine, CA, USA
6 Department of Biomedical Engineering, University of California, Irvine, Irvine, CA, USA
*Corresponding author. Tel: +1 949 824 2885; E-mail: marian.waterman@uci.edu
**Corresponding author. Tel: +1 949 824 8456; E-mail: jlowengr@uci.edu
†These authors contributed equally to this work

production) to favor glycolytic modes that produce lactate (Roche *et al*, 2001). Thus, at least in some tissues such as colon, Wnt signaling elevates PDK1 to suppress OXPHOS and to encourage glycolysis and the production of lactate.

Our previous study of xenograft colon tumors established that oncogenic Wnt signals directly activate PDK1 gene transcription as well as other glycolysis-connected gene targets including the lactate transporter MCT1 (*SLC16A1*) (Pate *et al*, 2014; Sprowl-Tanio *et al*, 2016). That Wnt signals might be directly responsible for shaping the metabolic profile of cells is a discovery from multiple studies focused on diseased [e.g., melanoma, breast (Sherwood, 2015)] and normal tissues (Esen *et al*, 2013). At least two types of Wnt signals have been defined. One signal utilizes canonical signaling and β-catenin-regulated transcription to drive sustained expression of glycolysis regulators. A second signal utilizes a novel Rac-mTORC2 pathway to increase the protein levels of glycolytic enzymes in the cytoplasm (Esen *et al*, 2013). Both signals can be triggered by secreted Wnt ligands, and these, in addition to oncogenic Wnt pathway activities created by genetic mutations, can direct the metabolic and proliferative capacity of colon tumors. However, because metabolism is shaped by the collective activity of multiple pathways and environmental influences—including those that enhance or diminish Wnt signaling—there is still much to learn about how signatures of metabolism are established.

Metabolic symbiosis has emerged as a powerful model to explain tumor heterogeneity and survival. As a concept, metabolic symbiosis means that glycolysis is not a singular metabolic choice for cells in a tumor; OXPHOS modes of metabolism may be dominant in subpopulations. The proposed outcome of this heterogeneity is that cooperation between two groups of cancer cells can maximize delivery and consumption of nutrients and minimize the environmental stresses that are imposed on a tumor. Glycolytic cells are likely the dominant consumers of glucose, and their fermentation of this carbon source produces an acidic by-product (lactate) that must be exported to the tumor microenvironment. Lactate can be angiogenic, and thus, the activities of glycolytic cells can be important for delivery of nutrients and growth factors to the tumor microenvironment (Murray & Wilson, 2001; Sonveaux *et al*, 2012). In turn, cancer cells with prominent modes of OXPHOS metabolism can uptake and utilize lactate (and other metabolic by-products) from neighboring glycolytic cells and metabolize it as a stable source of energy over long time scales (De Saedeleer *et al*, 2012; Epstein *et al*, 2014). An important example of this is the "reverse Warburg" effect observed in breast cancer (Martinez-Outschoorn *et al*, 2014). Thus, not all cancer cells show a preference for glycolysis at all times because microenvironmental, spatial, and temporal factors may direct them to emphasize OXPHOS modes of metabolism (Sonveaux *et al*, 2008; Pavlides *et al*, 2009; Obre & Rossignol, 2015). Such back-and-forth influences on glycolysis and OXPHOS create nongenetic tumor heterogeneity, meaning that genetically identical cancer cells might adopt different modes of metabolism depending on cell-intrinsic and cell-extrinsic influences. Identifying these influences and signals, and understanding the spatial and temporal forces that direct their cooperation is important, as metabolic symbiosis is not just a manifestation of tumor heterogeneity, but it is likely a fundamental aspect of tumor survival.

In the course of our study of Wnt signaling and glycolysis in xenograft colon tumors, we observed heterogeneous patterns of metabolism. Heterogeneity was observed via immunohistochemical stain of PDK1 activity, a major inhibitor of mitochondrial activity, and immunohistochemical stains of Wnt signaling. In particular, these stains revealed a pattern of discrete clusters of cells, or "spots", indicating groups of cells with different levels of glycolysis relative to OXPHOS, and differences in Wnt activity. We refer to these groups of cells as "glycolytic ($P_g$)" and "OXPHOS ($P_o$)" to indicate that they differ in the relative balance between these two modes of metabolism. As the spots of PDK activity and Wnt signaling appeared as a regular array in space, we hypothesized that metabolism was subject to rules of pattern formation, and we therefore developed a mathematical model with spatial features to study the organization of this pattern. Using reaction–diffusion equations to describe the dynamics of Wnt signaling, nutrients, cell substrates, and the populations of the different metabolic cell types, we elucidate the mechanisms that underlie this spatial pattern and find good agreement between the model and experiments. We lastly exploit this knowledge to identify promising therapeutic strategies.

# Results

### A spotted pattern of PDK activity and LEF-1 expression in xenograft tumors

Xenograft tumors of (human) colon cancer cell line SW480 (containing homozygous loss-of-function mutations in APC and intrinsically activated Wnt signaling) were produced by subcutaneous injection of cells in immunocompromised mice. To investigate metabolic changes within the tumor, 5.0- to 6.0-μm serial sections of formalin-fixed, paraffin-embedded tumor were probed with antisera specific for phosphorylated PDH (pPDH) as an indicator of PDK activity, and lymphoid enhancer factor-1 (LEF-1), a Wnt signaling transcription factor and Wnt target gene (Hovanes *et al*, 2001). Both stains revealed a general, high level of pPDH and LEF-1, but also heterogeneity in the form of a prominent spotted pattern (Fig 1A and B, where "mock" refers to tumors from parental SW480 cells). The pattern appeared as discrete localized clusters of cells with increased levels of pPDH, and these clusters, or spots, were detected at seemingly regular intervals. Since pPDH staining is an indicator of PDK activity, the darker stained cell clusters indicate increased rates of glycolysis relative to neighboring cells. The lighter staining, neighboring cells are likely utilizing greater levels of OXPHOS since PDH is less inhibited (less phosphorylated). Since it is known that lactate, the secreted by-product of glycolytic cells, can be imported into neighboring cells for use as an OXPHOS metabolic fuel, this pattern points to a potential symbiotic spatial relationship between these two cell populations, a metabolic relationship proposed by other groups studying cancer metabolism (Sonveaux *et al*, 2008; Pavlides *et al*, 2009)—glycolytic cells that are localized in distinct regions uptake glucose and produce metabolic fuel such as lactate for surrounding oxidative cells, a mode of sharing and metabolic distribution. In addition to the spotted metabolic pattern, an overlying gradient in pPDH staining level was observed wherein the spots were more densely arrayed toward the outer edges of the tumor, decreasing in frequency toward the center of the tumor, suggesting that more glycolysis occurs at the outer regions of the tumor where there is more vasculature (Pate *et al*, 2014; Appendix A1.4).

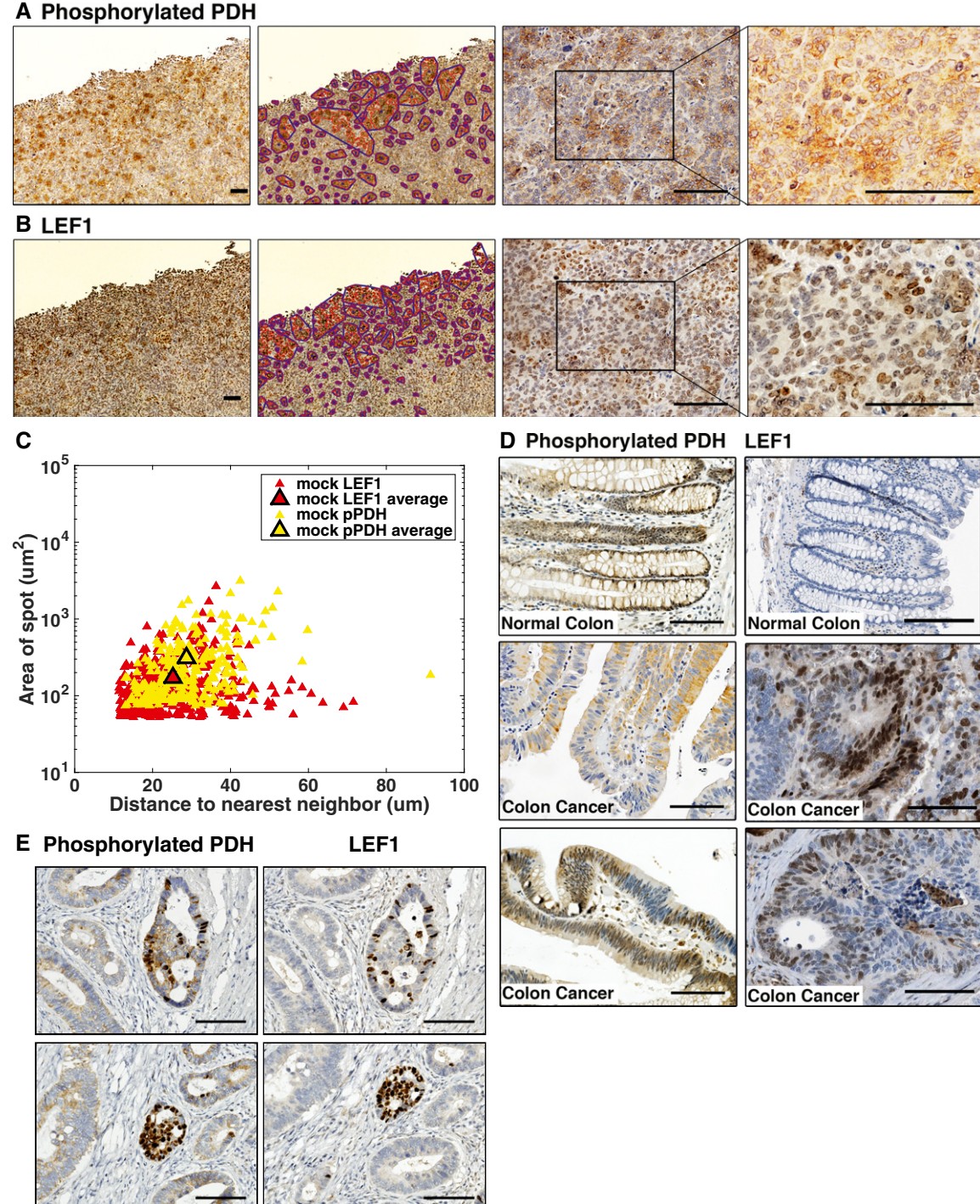

**Figure 1. SW480 xenograft tumors reveal a spotted pattern of metabolic heterogeneity.**

A, B    SW480 cells lentivirally transduced with empty pCDH vector (mock) were subcutaneously injected into immunocompromised mice. The resulting tumors were stained for (A) phosphorylated pyruvate dehydrogenase (pPDH) and counterstained with hematoxylin or (B) lymphoid enhancer factor-1 (LEF-1). Scale bars indicate 100 µm in the series of 4×, 20×, and 40× images. The red curves denote spot contours and the blue curves denote convex hulls, which group together spots that are sufficiently close to one another (see Appendix A1).

C    Image analysis of spot size versus distance of spot to nearest neighbor, using analyzed 20× images (third panels of A and B). The outlined data points indicate the average distance and area for pPDH and LEF-1 spots. Results show that quantifiable features of the spotted patterns in pPDH and LEF-1 are similar.

D    Colorectal carcinoma patient samples (tumors 1, 2, and 3) stained for pPDH (top) and LEF-1 (bottom) show spatial heterogeneity in expression levels. Scale bars are 100 µm (LEF-1 samples from Uhlén *et al*, 2015).

E    Serial section of human colorectal carcinoma stained with pPDH and LEF-1 antisera. Scale bars are 100 µm.

As we previously identified a link between Wnt signaling and glycolysis, we used immunohistochemistry to assess the activity of the canonical pathway. Interestingly, a spotted pattern was also evident in immunohistochemical stains of the Wnt target gene and effector, LEF-1 (Fig 1B), indicating that the spotted array might be linked to a pattern of heterogeneity in Wnt signaling. Automated image analysis was used to quantify the spatial parameters of each of the spotted patterns (see Appendix A1.1–A1.3 and Appendix Figs S1–S5). Figure 1C shows the quantification of each spot area and distances to each nearest neighbor, showing that the parameters of the spots for pPDH and LEF-1 are very similar (data on the number of cells per spot are given in Appendix A1.15). We found that the total area fraction of tumor covered by each set of spots in Fig 1A and B was nearly the same (pPDH: 21.2%; LEF-1: 20.2%). To assess the overlap between the pPDH and LEF-1 spots in the serial sections in Fig 1A and B (see Appendix A1.2), we counted spots that partially overlap and found that there was a significant overlap of 65–77% in the spatial arrangement of the pPDH and LEF-1 spots (Fig EV1). We also found that the area fraction of tumor covered by the overlapping region (pPDH spots that are contained in LEF-1 spots and LEF-1 spots that are contained in pPDH spots) is 7.4%.

To determine the significance of the association between the spots (see Appendix A1.2 for details), we analyzed staining in pairs of pixels, assuming that each pixel location in one section corresponds to the same pixel location in the other section. We performed a Cochran–Mantel–Haenszel test (Cochran, 1954; Mantel & Haenszel, 1959) and found that the pPDH and LEF-1 spots are significantly associated with one another ($P < 0.0001$). While this analysis is not definitive because it does not guarantee that the paired pixels are in the same cell (we found it difficult to directly match cells in the serial sections) and also does not take into account spatial variation in spot densities, it suggests that the patterned heterogeneity of metabolism and Wnt signaling are linked.

Xenograft tumors from colon cancer cell lines are different from primary human colon cancers, the latter of which develop in immunocompetent patients and contain a greater variety of cell types and stromal involvement. We asked whether PDK activity and Wnt signaling were uniform or heterogeneous in primary human colon tumors. In Fig 1D, pPDH and LEF-1 stains in primary human colon tumors compared to normal colon tissue demonstrate that there is indeed significant spatial heterogeneity in human tumors. In addition, serial sections of a primary human colon tumor stained with pPDH and LEF-1 antisera show a striking concordance in expression pattern (Fig 1E). While a regular spotted array is not apparent in primary tumors like it is in xenografts, the heterogeneous pattern of clusters of cells with high glycolysis and high LEF-1 in the epithelial portion of the tumor suggests that although xenograft tumors are artificial and have a different microenvironment, understanding the mechanisms underlying the observed spatial patterning in xenograft tumors can provide insight into the forces that create nongenetic heterogeneity in primary human colon tumors.

## Reaction–diffusion modeling mimics the self-organizing patterns of PDK activity and Wnt signaling in xenograft tumors

The regular spotted pattern in the xenograft IHC stains suggests the development of a mathematical model consisting of reaction– diffusion equations similar to those first described by Alan Turing (Turing, 1952). Turing's equations describe how an initial perturbation in the concentrations of chemicals, or morphogens, can grow in the presence of diffusion (the Turing instability) and self-organize into a spatial pattern. Because diffusion is normally a stabilizing process, diffusion-driven instabilities occur only under certain conditions (Murray, 2003; Kondo & Miura, 2010). Recently, Marcon *et al* (2016) performed an automated analysis of Turing-type reaction–diffusion equations and identified general conditions for which instabilities could occur. When two species are considered (e.g., activator–inhibitor models), the species need to diffuse at sufficiently different rates as observed previously (e.g., short-range activator, long-range inhibitor). However, when multiple diffusing species are present, instabilities can be obtained even for arbitrary diffusivities. Here, we focus on reaction–diffusion models that link cell metabolic phenotypes with Wnt signaling and argue that conditions for instability are met in colon cancer.

Despite the fact that colon cancers are most often driven by genetically activated Wnt signaling, a cell-autonomous condition, there are numerous studies that highlight that secreted Wnt ligands and their bona fide signaling through Frizzled receptors on the plasma membrane are abundantly active in human colon cancer and that they influence colon cancer biology (Holcombe *et al*, 2002; Seshagiri *et al*, 2012; Voloshanenko *et al*, 2013; Giannakis *et al*, 2014). Importantly, Wnt ligands are highly constrained in their diffusion, traveling only one to two cells from the origin of their secretion, meaning that the range of their influence is highly localized (Farin *et al*, 2016). This is in contrast to the longer-range diffusion properties of known, secreted inhibitors that bind to Wnt ligands and/or interfere with receptor binding (i.e., DKK, SFRPs) (Mii & Taira, 2009, 2011). Some of these inhibitors are Wnt target genes, for example, DKK4, an inhibitor that is expressed in human colon cancer, and SFRP2, a secreted Wnt inhibitor induced by Wnt4 in the developing kidney (Lescher *et al*, 1998). Thus, a Turing-type model, wherein short-range nonlinear activation by Wnt ligands and long-range inhibition of their activities, fits well with the known physical and regulatory properties of Wnts and their inhibitors. Moreover, this type of model is capable of forming patterns (Murray, 2003; Kondo & Miura, 2010). Previously, Turing models have been used to describe Wnt-directed patterns in a variety of contexts including hair follicles (Sick *et al*, 2006; Kondo & Miura, 2010), colon crypts (Zhang *et al*, 2012), limb development (Raspopovic *et al*, 2014), and stem cell-driven cancers (Youssefpour *et al*, 2012; Yan *et al*, 2016). Additionally, the BMP family, known to be Wnt signaling antagonists, has been recently described to direct murine intestinal patterning (Walton *et al*, 2015).

We therefore developed a Turing-type model for simulating the spatial and temporal dynamics of different metabolic phenotypes, nutrients, and Wnt signaling activity through a system of reaction– diffusion equations (Fig 2A and B; Appendix A2 and A3). We included populations of cells that perform less glycolysis, which we refer to as oxidative ($P_o$) cells, and those that perform more glycolysis, which are termed glycolytic ($P_g$) cells. Both types of cells may divide, die, and undergo random movement. Depending on local environmental conditions, the cells may switch from one phenotype to the other. A diffusible substrate (N), which accounts for concentrations of nutrients such as glucose and growth factors, regulates cell division and death ($\chi_N$), the switching function $\chi_W^*$ $\chi_N^*$ from

OXPHOS to glycolysis, and the ability of cells to generate Wnt (W) and Wnt inhibitor ($W_I$) activities. The Wnt and Wnt inhibitor equations are based on the Gierer–Meinhardt activator–inhibitor model (Gierer & Meinhardt, 1972), where Wnt is the short-range activator which produces a long-range factor that inhibits Wnt activity (e.g., SFRP2). Because Wnt signaling is assumed to be constitutively active, both OXPHOS and glycolytic cells are assumed to upregulate Wnt activity at the rate $S_W$. In the model shown in Fig 2A and B, the glycolytic cell proliferation rates and the metabolic switching rates ($\chi_W$ and $\chi_W^*$) also depend on Wnt activity where a higher activity level increases cell propensities for glycolysis over OXPHOS, if sufficient nutrients are available ($\chi_N^*$). To model the angiogenic response of the mouse vasculature to the lactate produced by the glycolytic cells and the accompanying increased delivery of nutrients, we introduced sources $N_S$ that increase the amount of nutrient in the system proportionally to the amount of glycolytic activity of the cells. We also assumed that the vascular density was largest at the domain boundary and thus, we modified the boundary conditions for nutrients analogously. See Appendix A2 for the precise functional relationships.

We also considered a more general *in vivo* model, which accounted for PDK activity, hypoxia-inducible transcription factor concentrations (HIF1α), lactate concentration, and cross-feeding between glycolytic and OXPHOS cells (Appendix A3). Assuming that Wnt and HIFs promote PDK expression/activity (Kim *et al*, 2006; Pate *et al*, 2014; Prigione *et al*, 2014), that PDK activity promotes lactate production (Pate *et al*, 2014), and that lactate increases HIF1α expression levels and provides a source of fuel for OXPHOS cells (De Saedeleer *et al*, 2012; Epstein *et al*, 2014), we obtained results that were qualitatively similar to the simpler model shown in Fig 2A and B where these additional processes were not considered directly. In particular, the effects of Wnt signaling dominate those of cross-feeding between the cell types, and the positive feedback loop between Wnt and PDK (high PDK implies more $P_g$ cells; higher numbers of $P_g$ cells imply more Wnt activity; more Wnt activity means increased PDK) has been distilled in the simpler model so that Wnt activity level, rather than PDK levels, provides an effective metabolic switch between relative amounts of OXPHOS and glycolysis. Because PDK drives the switch in metabolism in SW480 cells, we use the $P_g$ and $P_o$ spatial distributions to compare to our xenograft stains.

The model equations were solved in nondimensional form using a characteristic proliferation time $T$ of 1 day to rescale time and a characteristic diffusion length $l$ of the Wnt inhibitor to rescale space.

Since we did not know $l$ (in fact, there may be many factors that contribute to Wnt inhibition), we varied $l$ and found good agreement between the experimental and numerical patterns when $l \approx 40\ \mu m$. A full description of the both models, boundary conditions, and the nondimensionalization can be found in Materials and Methods and in the Appendix. The parameter values can be found in Table 1 (Jiang *et al*, 2005; Rockne *et al*, 2010; Mendoza-Juez *et al*, 2012).

Figure 2C presents the numerical results for the fractions of glycolytic and oxidative cells and the concentrations of Wnt and Wnt inhibitor, where each two-dimensional plot is a horizontal slice through the center of the three-dimensional spatial domain (nutrient distributions can be found in Appendix Fig S15 and Appendix A4). The cells were initially seeded randomly near the boundary of the domain to reflect the fact that the cells that survive the initial implantation are likely close to nutrient sources (alternative initial distributions of cells give similar results). The cells then proliferate and grow inwards toward the center of the domain with angiogenesis-induced nutrient sources fueling the growth. Consistent with the xenograft data, a distinct spotted pattern in the population of glycolytic cells is produced by the model over time. Over the entire domain, there is a high level of glycolytic-dominant cells with localized areas of highly active glycolytic cells (dark red spots). Similar to the xenograft tumors, the spots are denser toward the boundary of the tumor space, where there is a higher density of vasculature, a spatial pattern that agrees with the overall pattern of pPDH staining of the mock tumors in Fig 1A. The oxidative cell fractions are close to 0 in the same spots where glycolysis is high, and their levels are relatively higher in regions surrounding these spots. Wnt and Wnt inhibitor activity show a similar pattern, with high levels distributed in a spotted array throughout the domain, surrounded by lower levels in the neighboring regions. Like the pattern of glycolysis, the frequency of spots is higher near the boundary relative to the interior. The square symmetries in the simulated spot distributions are due to the use of a cubic spatial domain in the simulations. Quantitative and comparative analysis of the patterns in the xenograft tumors to the simulated pattern generated by the model indicates that the model predicts similar dimensions for the size of the spots and distance between the spots (see Fig 3D and E), although there is significant scatter in the data.

Because the model parameters were largely unknown, we investigated their influence on the results through a parameter study (Appendix A5). Using a diffusive stability analysis to determine the ranges of values for which patterns were predicted to occur (see

**Figure 2.  A mathematical model for Wnt signaling regulation of metabolism.**
This set of reaction–diffusion equations describes the change over time of oxidative ($P_o$) and glycolytic ($P_g$) cell populations, Wnt signaling activity (W), and Wnt inhibitor activity (WI).

A   The cells can diffuse, proliferate, and "switch" metabolism programs depending on Wnt signaling activity and nutrient levels and die from lack of nutrient (N).

B   Wnt and Wnt inhibitor activity equations are based on the Gierer–Meinhardt activator–inhibitor model. The Wnt signal diffuses short range relative to the longer-range diffusion of the Wnt inhibitor. Wnt also auto-upregulates its activity in glycolytic cells at a rate proportional to nutrient level, is inhibited by a Wnt inhibitor, is constitutively upregulated in both cell types, and decays (downregulation term). The Wnt inhibitor diffuses long range, is nonlinearly upregulated by Wnt, and decays. Equations for nutrient and dead cells ($P_d$) are not shown; their descriptions are in the main text.

C   Three-dimensional numerical simulations that model the spatial distribution and level of glycolytic and oxidative cells, Wnt, and Wnt inhibitor reveal an emergent self-organizing pattern of metabolic heterogeneity (spots). The simulations shown depict the heterogeneity in a 3D and 2D representation. The 3D representation includes a portion of the "tumor" removed to visualize the interior of the domain. The 2D representation is a horizontal slice of the respective 3D simulation in the center of the domain. Color bars refer to unitless concentrations.

D   Summary of parameter effects on the spotted pattern.

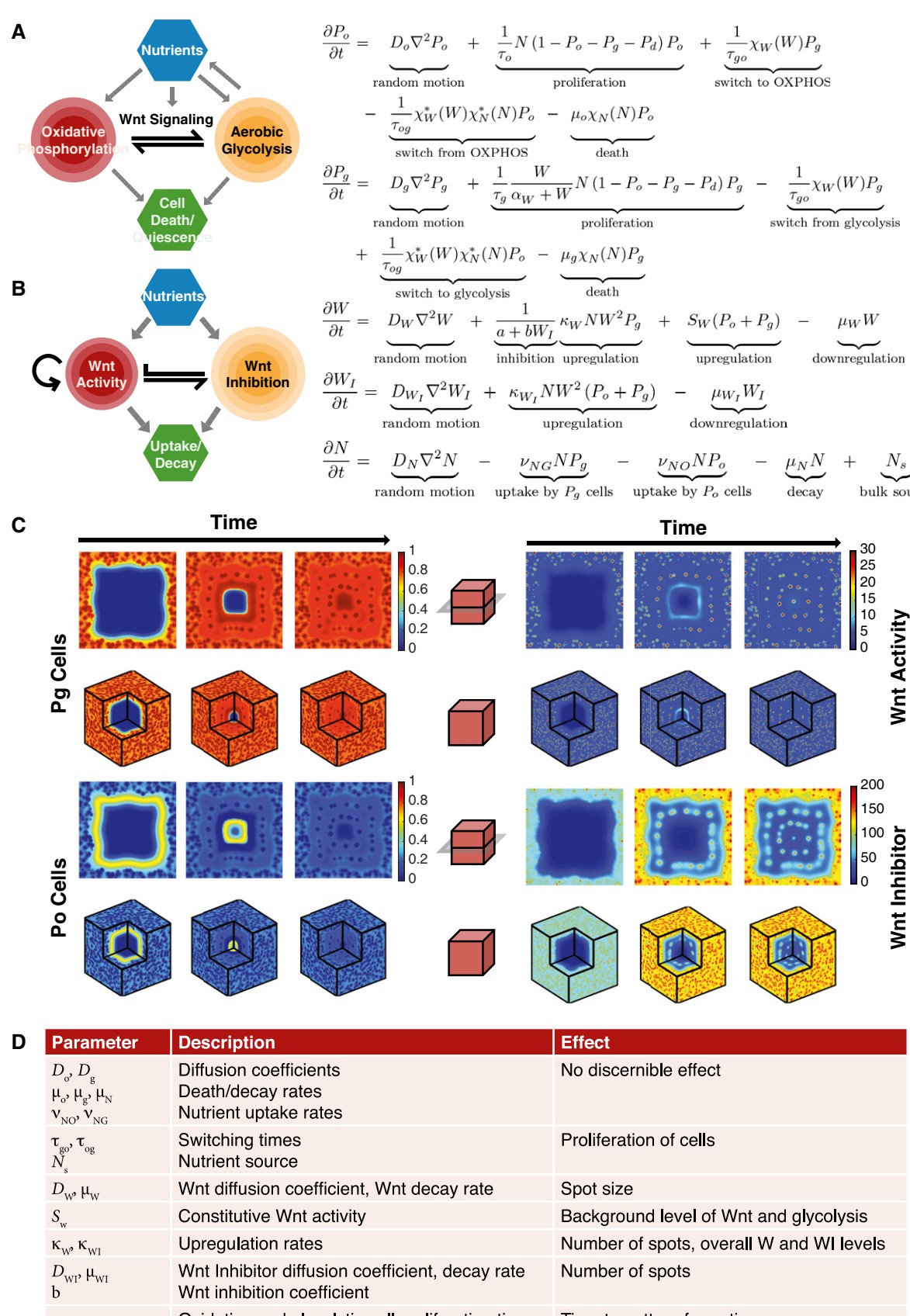

| Parameter | Description | Effect |
|---|---|---|
| $D_o$, $D_g$<br>$\mu_o$, $\mu_g$, $\mu_N$<br>$\nu_{NO}$, $\nu_{NG}$ | Diffusion coefficients<br>Death/decay rates<br>Nutrient uptake rates | No discernible effect |
| $\tau_{go}$, $\tau_{og}$<br>$N_s$ | Switching times<br>Nutrient source | Proliferation of cells |
| $D_W$, $\mu_W$ | Wnt diffusion coefficient, Wnt decay rate | Spot size |
| $S_w$ | Constitutive Wnt activity | Background level of Wnt and glycolysis |
| $\kappa_W$, $\kappa_{WI}$ | Upregulation rates | Number of spots, overall W and WI levels |
| $D_{WI}$, $\mu_{WI}$<br>$b$ | Wnt Inhibitor diffusion coefficient, decay rate<br>Wnt inhibition coefficient | Number of spots |
| $\tau_o$, $\tau_g$ | Oxidative and glycolytic cell proliferation time | Time to pattern formation |

Figure 2.

**Table 1. Parameters.**

| Parameter | Description | Mock value | dnLEF value | References |
|---|---|---|---|---|
| $D_o$ | Diffusion coefficient of oxidative cells | 0.01 | 0.01 | Rockne *et al* (2010) |
| $D_g$ | Diffusion coefficient of glycolytic cells | 0.01 | 0.01 | Rockne *et al* (2010) |
| $D_W$ | Diffusion coefficient of Wnt | 0.004 | 0.008 | Chosen to be small |
| $D_{W_I}$ | Diffusion coefficient of Wnt inhibitor | 1 | 1.5 | Chosen to be large |
| $D_N$ | Diffusion coefficient of nutrient | 100 | 100 | Jiang *et al* (2005) |
| $\tau_o$ | Oxidative cell proliferation time | 1 | 1 | Non-dimensionalization (time scale) |
| $\tau_g$ | Glycolytic cell proliferation time | 1 | 1 | Non-dimensionalization (time scale) |
| $\tau_{og}$ | Switch time from OXPHOS to glycolysis | 1/24 | 1/24 | Mendoza-Juez *et al* (2012) |
| $\tau_{go}$ | Switch time from glycolysis to OXPHOS | 1 | 1 | Mendoza-Juez *et al* (2012) |
| $\alpha_W$ | Constant for Michaelis-Menten dynamics | 1 | 1 | Parameter estimation |
| $\kappa_W$ | Rate of nonlinear Wnt production | 5 | 5 | Parameter estimation |
| $\kappa_{WI}$ | Rate of Wnt inhibitor production | 1 | 1 | Parameter estimation |
| $\mu_o$ | Decay rate of $P_o$ cells | 1 | 1 | Parameter estimation |
| $\mu_g$ | Decay rate of $P_g$ cells | 1 | 1 | Parameter estimation |
| $\mu_d$ | Decay rate of $P_d$ cells | 1 | 1 | Parameter estimation |
| $\mu_W$ | Decay rate of Wnt | 2 | 2 | Parameter estimation |
| $\mu_{W_I}$ | Decay rate of Wnt inhibitor | 3 | 3 | Parameter estimation |
| $\mu_N$ | Decay rate of nutrient | 0.1 | 0.1 | Parameter estimation |
| $S_W$ | Rate of Wnt production through cells | 7.5 | 6.5 | Parameter estimation |
| $a$ | Constant of inhibition | $10^{-8}$ | $10^{-8}$ | Parameter estimation |
| $b$ | Constant of inhibition by $W_I$ | 1 | 1 | Parameter estimation |
| $\gamma_W$ | Sensitivity level of Wnt switch functions | 1 | 1 | Parameter estimation |
| $\gamma_N$ | Sensitivity level of nutrient switch function | 100 | 100 | Assumed to be high |
| $\nu_{NG}$ | Uptake of nutrient by $P_g$ cells | 10 | 10 | Parameter estimation |
| $\nu_{NO}$ | Uptake of nutrient by $P_o$ cells | 10 | 10 | Parameter estimation |
| $N_s$ | Parameter for nutrient source | 2 | 2 | Parameter estimation |
| $W^*$ | Wnt level at which 50% of cells switch metabolism | 5 | 5 | Parameter estimation |
| $N^*$ | Nutrient level below which cells die | 0.07 | 0.07 | Parameter estimation |
| $N_g^*$ | Nutrient level below which $P_o$ cells cannot switch to glycolysis | 0.1 | 0.1 | Parameter estimation |
| $\alpha_N$ | Value of scaling function when $\int P_g = 0$ | 0.025 | 0.025 | Parameter estimation |
| $S_x$ | Horizontal length of spatial domain | 12 | 12 | |
| $S_y$ | Vertical length of spatial domain | 12 | 12 | |

Model parameters for mock and dnLEF/dnTCF simulations.

Appendix A6), we modified the parameters one by one within the pattern-forming range and tested for phenotype changes in metabolism and patterning. The results are summarized in the table in Fig 2D. Increasing the Wnt diffusion coefficient or decreasing the Wnt decay rate increased the extent of Wnt activity, so that the spots of glycolysis increased in size. Increasing the Wnt inhibitor diffusion coefficient or decreasing the decay of the Wnt inhibitor caused the inhibitor to stay within the system for longer times and resulted in fewer spots. Modifying the switching times between the phenotypes changed the proportion of $P_o$ and $P_g$. Increasing or decreasing the Wnt switch changed the background levels of $P_g$ cells and spot sizes without affecting the number of spots. Small reductions in $S_W$, which can be thought of as reducing overall Wnt signaling, reduced the background levels of glycolytic cells without much effect on the sizes or numbers of spots. Sufficiently reducing $S_W$ resulted in all terms ($P_o$, $P_g$, $W$, and $W_I$) decreasing to 0. Decreasing $\kappa_W$ (nonlinear Wnt activity) or increasing $b$ (Wnt response to inhibition) paradoxically increases the number of glycolytic cells because nonlinear interactions actually result in a decreased amount of $W_I$. Analogously, when $\kappa_{WI}$ (nonlinear Wnt inhibitor activity) decreases, the number of glycolytic cells decreases. Modifying the cell diffusion coefficients, death and decay rates, and the nutrient uptake rates did not significantly influence the self-organization of a spotted array. Similarly, varying the proliferation times only changed the time it took to reach a steady state but otherwise had no effect on pattern formation.

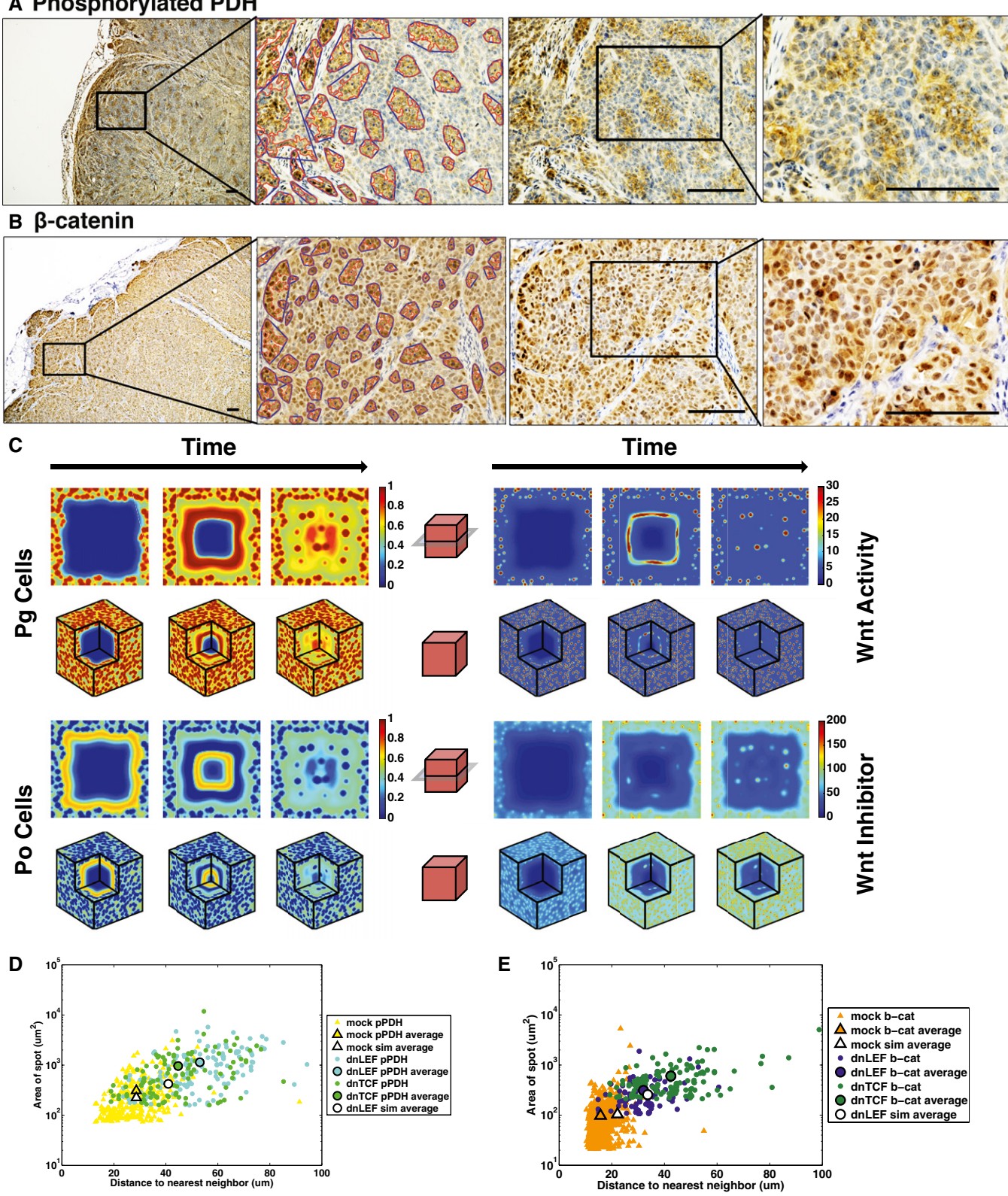

Figure 3.

**Figure 3.  Decreasing Wnt signaling leads to changes in metabolic patterning in xenograft tumors.**

A, B   SW480 cells were lentivirally transduced to express dominant negative LEF-1 (dnLEF-1), and transduced cells were injected subcutaneously into immunocompromised mice. Tumor sections were stained for phosphorylated PDH (A) and β-catenin (B) and counterstained with hematoxylin. Compared to mock tumors, the spots are larger and more heterogeneous and the background staining is lighter, which reflects an overall decrease in Wnt signaling. Scale bars are 100 μm. The red curves denote spot contours and the blue curves denote convex hulls, which group spots that are sufficiently close to one another (see Appendix A1).

C   Numerical simulations that lower Wnt signaling activity in the model show an overall decrease in glycolysis and a change in the spotted pattern that closely mimics that observed in the dnLEF tumors. Color bars refer to unitless concentrations.

D   Image analysis of spot size versus distance of spot to nearest neighbor, using analyzed images. Averages for mock and dnLEF spot simulations are denoted in white outlined symbols (pPDH: spot sizes/inter-spot distances: mock simulation average: 225 ± 278 μm²/29 ± 12 μm; mock xenograft tumor average: 309 ± 367 μm²/29 ± 10 μm; dnLEF-1 simulation average: 423 ± 327 μm²/41 ± 14 μm; dnLEF-1 xenograft tumor average: 1,139 ± 1,042 μm²/53 ± 15 μm). Results are also shown for dominant negative transcription factor 1 (dnTCF-1) tumors (see Appendix A1.10–A1.12 and Appendix Figs S8–S10). The metabolic pattern in dnTCF-1 tumors is consistent with that in dnLEF-1 tumors. The analysis and model predict that the changes in the metabolic spotted pattern (larger spots, greater distance between spots) are due to an increase in the diffusion of Wnt and the Wnt inhibitor.

E   Comparison of mock β-catenin spots to dnLEF-1 and dnTCF-1 β-catenin spots from image analysis. Averages for mock and dnLEF-1 spot simulations are denoted in white outlined symbols (β-catenin spot sizes/inter-spot distances: mock simulation average: 139 ± 145 μm²/26 ± 11 μm; mock xenograft tumor average: 97 ± 209 μm²/16 ± 4 μm; dnLEF-1 simulation average: 342 ± 221 μm²/39 ± 14 μm; dnLEF-1 xenograft tumor average: 312 ± 277 μm²/32 ± 9 μm; dnTCF-1 xenograft tumor average: 603 ± 578 μm²/42 ± 14 μm). The analysis and model predict that Wnt signaling diffuses further with dominant negative LEF-1 expression.

## Interfering with Wnt signaling alters colon cancer metabolic patterns *in vivo*

Since our model utilizes Wnt signaling, we tested how interference of this pathway would alter metabolic patterning. To disrupt the pathway, we used lentiviral transduction to express dominant negative LEF-1 (dnLEF-1) or dominant negative TCF-1 (dnTCF-1) transcription factors. Both dominant negative versions are naturally occurring LEF/TCF isoforms that lack the β-catenin binding domain and therefore interfere with the activation/expression of Wnt target genes. Expression of moderate, physiological levels of dnLEF-1 or dnTCF-1 expression, partially, but not completely, disrupts Wnt target gene expression in the xenograft tumors (Van de Wetering *et al*, 2002; Hoverter *et al*, 2012; Pate *et al*, 2014). Partial disruption is necessary because complete inhibition of Wnt activity would block cell cycle progression and the formation of tumors altogether.

SW480 colon cancer cells that had been lentivirally transduced and selected for dnLEF-1 or dnTCF-1 expression were subcutaneously injected into immunocompromised mice for tumor formation. Experiments showed that, as a result of dnLEF-1 expression, PDK1 activity was reduced, Warburg metabolism was diminished, and tumor mass was reduced approximately four- to fivefold (Pate *et al*, 2014). Immunohistochemical staining of the levels of phospho-PDH in these tumors (Fig 3A) revealed a lighter background and lower pPDH level overall. Interestingly, pPDH positivity remained easily visible in clusters of cells, but there were striking changes in the spotted pattern. Each pPDH-positive cluster was comprised of a larger number of cells (mock average: ~seven cells/spot and dnLEF average: ~17 cells/spot; see Appendix A1.15 and Appendix Fig S11), and there was a greater distance between each spot (compared to parental, mock-transduced cells; Fig 3D). We also utilized immunohistochemical staining for the Wnt-mediating factor β-catenin in the dnLEF tumors (Fig 3E) (dnLEF-expressing tumors cannot be stained for LEF-1). These stains revealed a spotted pattern, with clusters of cells having higher levels of β-catenin in the nucleus than neighboring cells, although because of the very high levels of β-catenin in SW480 cells, there was an overall strong intensity of the IHC stain. Image analysis showed that while the β-catenin$^{HI}$ spots are, on average, smaller than the pPDH-positive spots, they too had increased in size and distance relative to the pattern of β-catenin$^{HI}$ cell clusters in the mock/parental tumors.

Additional image analyses of staining for pPDH and β-catenin in dnLEF-1- and dnTCF-1-expressing tumors are provided in Appendix A1 (Appendix A1.8–A1.12), together with a quantification of these staining patterns (Appendix A1 and Appendix A1.13–A.1.15). In summary, there were significant changes in both the intensity and distribution of the spotted patterns for pPDH and β-catenin when Wnt signaling was reduced by dnLEF-1/dnTCF-1 expression.

## Reaction–diffusion modeling of metabolic patterns under partial disruption of Wnt signaling predicts expression of factors that increase the range of Wnt signaling

To understand the phenotypic changes in the spotted patterns when Wnt signaling was partially disrupted, we used our model to identify changes in parameters that could recapitulate the experimental observations. The simplest change was to reduce $S_W$, which mimics dnLEF-1 and dnTCF-1 expression in lowering intrinsic Wnt activity throughout the domain, a manipulation that represents the cell-autonomous effect of expressing Wnt-interfering, dominant negative LEF/TCF factors in the nucleus of every cell. However, as described earlier in our parameter study, decreasing $S_w$ lowers the overall background levels of $P_g$ cells, but does not affect the spotted pattern (unless it is taken to be too small in which case the pattern disappears). Thus, solely lowering overall Wnt signaling ($S_W$) in the model produces outcomes in pattern that are inconsistent with the experimental data.

Clearly, the effects of dnLEF-1/dnTCF-1 expression are more complex than the cell-autonomous manipulation of only decreasing Wnt pathway activity in the nucleus. We considered the possibility that dnLEF-1/dnTCF-1 might also be triggering a cell-extrinsic response that connects collections of cells in the microenvironment. Specifically, our parameter study suggested that the increase in the sizes of pPDH-positive cell clusters might be due to extracellular soluble factors that increase the range of the activator (Wnt ligands) and that the decrease in the number of pPDH-positive cell clusters could be due to factors that increase the range of inhibition. Therefore, we included two additional parameter modifications: increasing $D_W$, which increases the range of Wnt ligands and makes the spots larger, and concomitantly increasing $D_{WI}$, which increases the range of Wnt ligand inhibitors and reduces the number of spots.

Changing these two parameters and decreasing $S_w$ simultaneously resulted in a striking recapitulation of the changes in the spotted pattern observed in the dnLEF-1/dnTCF-1-expressing tumors: lower background levels of $P_g$ cells and larger, fewer spots of $P_g$-glycolysis (Fig 3C). The average sizes and centroid distances of the $P_g$ spots in the simulated tumors correlated very well with the experimental observations (Fig 3D). Further, the simulation showed a decrease in nutrient concentrations throughout the tumor (Appendix A4), a result that is consistent with our previous experimental data as we observe significantly fewer blood vessels in the dnLEF-1 and dnTCF-1 tumors (Pate *et al*, 2014). This is because the nutrient concentration $N$ is linked to the proportion of $P_g$ cells, which are decreasing.

Since in the experiments, we used IHC staining of β-catenin as a direct assessment of patterns in Wnt signaling, in the simulations, we analogously examined patterns of Wnt activity in the model. The results show very good agreement between the simulations and the experiments: The spots of Wnt activity are smaller than the $P_g$ spots but the Wnt-activity spots were increased in size and distance relative to the pattern of Wnt activity in the simulations of the mock tumors (Fig 3E). In summary, our results suggest that stressing the colon cancer cells by interfering with Wnt signaling triggers changes in the expression of factors that increase the diffusion range, or "spread", of Wnt ligands and extend the range of Wnt inhibition.

### *In vivo* validation of model predictions

Only a few studies have directly examined the diffusion range of Wnt ligands in any tissue, a range which is extremely limited, in part because the ligands are post-translationally modified by palmitoylation and are highly lipophilic for membranes and extracellular matrix proteins (Willert *et al*, 2003; Farin *et al*, 2016). There is a growing awareness of proteins that modify the range of ligand diffusion, although their actions and impact are not very well characterized (Fig 4A). Perhaps the best-characterized factors that influence Wnt ligand diffusion are the SFRP protein family, secreted inhibitors that bind directly to Wnt ligands and interfere with receptor binding. Importantly, several studies have shown that even though SFRPs can interfere with Frizzled receptor binding, they are bimodal in their actions, repressing Wnts at high concentrations of ligand but also promoting Wnt actions by increasing their range of diffusion and, in essence, delivering the ligands to cells that are further away (Mii & Taira, 2009, 2011). Given that our mathematical model predicts the diffusion of Wnt ligands and their inhibitors have increased in the dnLEF-1 and dnTCF-1 xenograft tumors, we tested the prediction that one or more candidate regulators of Wnt diffusion were elevated in their expression. Using RNA-seq data as a guide for identifying candidates expressed in SW480 cells, we designed human-specific primers for both diffusion regulators and inhibitors that were detectably expressed in this cell line. Expression analysis of mRNA purified from 2D cultures and 3D xenograft tumors revealed that the Wnt diffusers SPOCK2, GPC4, and SFRP5 are upregulated specifically in dnLEF-1 and dnTCF-1 xenograft tumors but not 2D culture (Fig 4B and C). Since the primers are human specific, the expression changes derive specifically from the human cancer cells and not mouse-derived cell types in the tumor microenvironment.

While small-molecule Wnt inhibitors that mimic the effects of dnLEF-1 and dnTCF-1 are working their way through pre-clinical testing and early-phase clinical trials, there are not yet any available data from patient studies that profile gene expression changes in primary colorectal cancers treated with Wnt inhibitors. However, there are limited data available from patients treated with radio- and chemotherapy regimens, treatments that induce stress and loss of nutrient delivery to the tumor. We analyzed one dataset [(Snipstad *et al*, 2010) NCBI GEO GDS3756], which provided gene expression profiles of a group of colorectal cancer patients before and after radio- and chemotherapy treatment. Figure 4D and E shows that, while the treatment had no significant effect on expression of Wnt ligand regulators in normal rectal tissue, the expression of GPC1 and three SFRP family members (SFRP1, SFRP2, and SFRP4) was strongly and specifically increased in the tumor following treatment (Fig 4E). We checked for changes in expression of Wnt ligands, and although there was a trend toward significantly increased expression of Wnt2, Wnt5b, Wnt8b, and Wnt10b specifically in the tumor and not the neighboring normal tissue, the changes did not quite reach statistical significance (Fig EV2A and B). Interestingly, one glycolytic gene (ENO2) was significantly increased in radiochemotherapy-treated tumor tissue (Fig EV2D), and the glycolytic regulator HIF1A was increased but not to the same level of significance. This suggests that radiochemotherapy may trigger increased expression of proteins that increase the range of Wnt diffusion, a response that we predict might serve to maintain a critical level of glycolytic cells in the tumor.

### Modeling a therapeutic treatment for cancer: metabolic targeting

To test whether glycolytic cells are the important subpopulation of cells to target in the tumors, we compared the effectiveness of a hypothetical therapy program that selectively targeted each population by independently varying the death rates of $P_o$ and $P_g$ cells (we introduced additional death terms $\mu_{P_o} P_o$ and $\mu_{P_g} P_g$ in the $P_o$ and $P_g$ equations in Fig 2A). The simulation applied the targeted therapy to a fully developed tumor at steady state for different lengths of time (days), followed by removal of the therapy and a recovery time for tumor development (Fig 5). In this figure, the tumor size (integral of $P_g + P_o$ over the entire domain) is shown relative to that of the untreated tumor (see Fig EV3 for the dynamics of the individual cell populations). The treatment dose refers to cell death rates $\mu_{P_o}$ or $\mu_{P_g}$, and targeting means that the death rate is nonzero only for the target cell population. These simulations revealed that regardless of the targeted population, modest rates of cell death suppressed tumor development transiently, followed by full recovery of the system once therapy was removed, a pattern more evident and more robust when cell killing was directed toward the $P_o$ population. At sufficiently large death rates, complete loss of the tumor could be achieved. However, targeting the $P_g$ population led to a complete loss of the simulated tumor at shorter treatment times and smaller death rates than when $P_o$ cells were targeted. Thus, the simulation predicts that $P_g$ cells are the more sensitive population and that targeting these cells could more effectively lead to a full regression of the tumor.

Since selective targeting resulted in full recovery of the simulated tumors unless death rates were sufficiently high and treatment was sufficiently long, we considered dual targeting of two features of cancer cell metabolism as a mechanism for more effective killing.

**A**

| Fly Gene | Human Gene | Reference | Function |
|---|---|---|---|
| | CDC42 | Stanganello, et al. *Nature Communications* 2015 | CDC42/N-Wasp regulates formation of Wnt-positive filopodia |
| Swim | LCN7/TINAGL1 | Mulligan et al. *PNAS* 2012 | Binds to Wg in a lipid-dependent manner to increase range. At high concentrations, can inhibit Wls activity. |
| Cow | Testican-2/ SPOCK2 | Chang, et al. *PloS One* 2014 | HSPG that binds to Wg, increasing Wg mobility. Can act as an inhibitor in high Wg concentration. |
| Dally/Dlp | GPC1-6 | Lin, et al. *Nature* 1999 | Co-receptor with Fzd2 to modulate range of Wg |
| | SFRP1-5 | Mii & Taira. *Development.* 2009; Mii & Taira. *Dev. Growth & Diff.* 2011 | Bimodal actions: repress Wnt by inhibiting Wnt ligand:Fzd receptor binding; promote Wnt by increasing range of ligand diffusion |

**B**

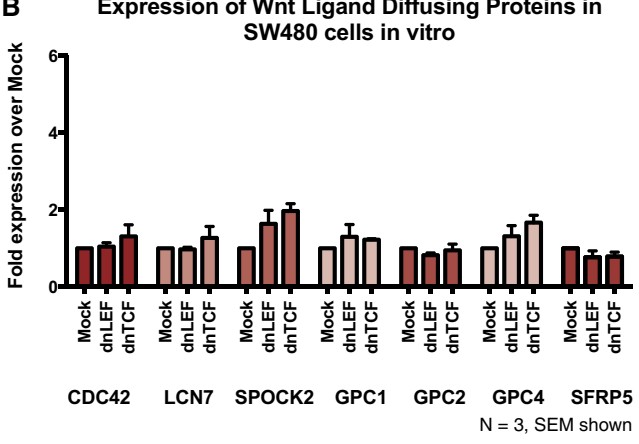

**C**

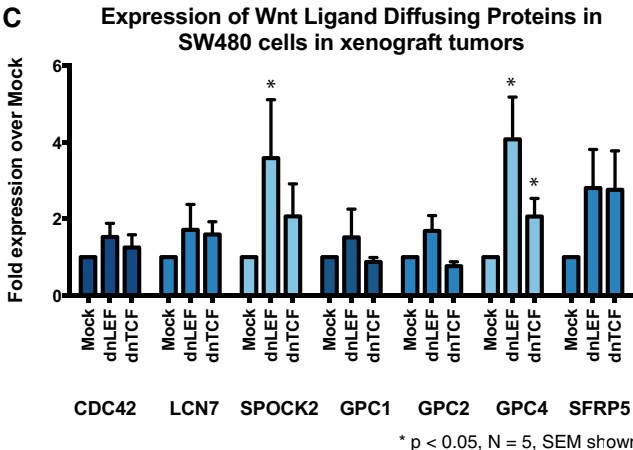

**D**

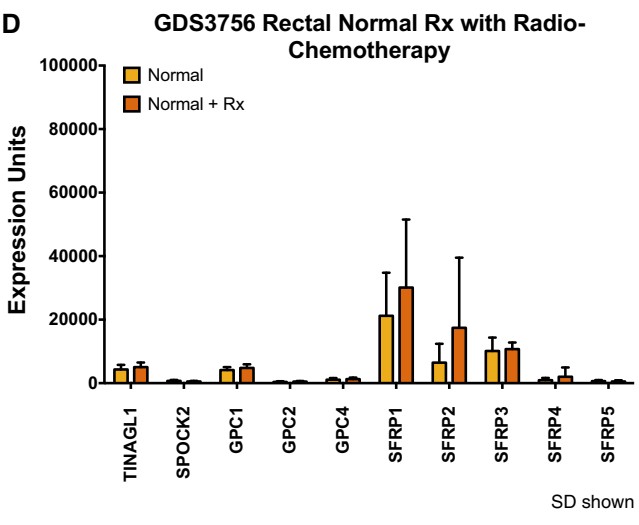

**E**

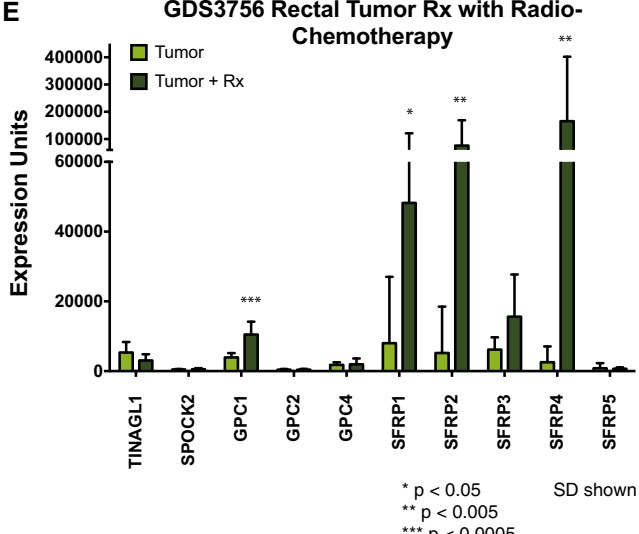

**Figure 4. Model predictions revealed in xenograft tumors and human colorectal cancer.**

The model predicted that lowering Wnt signaling results in an increase in the expression of factors that increase the range of diffusion of Wnt and Wnt inhibitors.

A    Known regulators of Wnt ligand diffusion.

B, C    Quantitative PCR of diffusion regulators in SW480 mock, dnLEF-1, and dnTCF-1 (B) transduced cells, and (C) xenograft tumors show human SPOCK2, GPC4, and SFRP5 mRNA are notably upregulated in xenograft tumors but not 2D *in vitro* culture. *In vitro* data represent an average of three sample sets ($\pm$ SEM), and xenograft tumor data represent the average of five independent tumor sets ($\pm$ SEM); * denotes $P < 0.05$. Statistical significance was determined using Student's two-tailed *t*-test.

D, E    Gene expression data, from GEO dataset GDS3756, of 21 rectal cancer patient tissue with or without radio-chemotherapy (Snipstad *et al*, 2010). Significant changes in expression levels of GPC1 ($P = 0.00019$), SFRP1 ($P = 0.016$), SFRP2 ($P = 0.0006$), and SFRP4 ($P = 0.0006$) are observed in post-therapy tumor cells compared to before treatment. SD shown. Statistical significance was determined using the Mann–Whitney *U*-test with Benjamini–Hochberg correction for multiple hypotheses (RStudio).

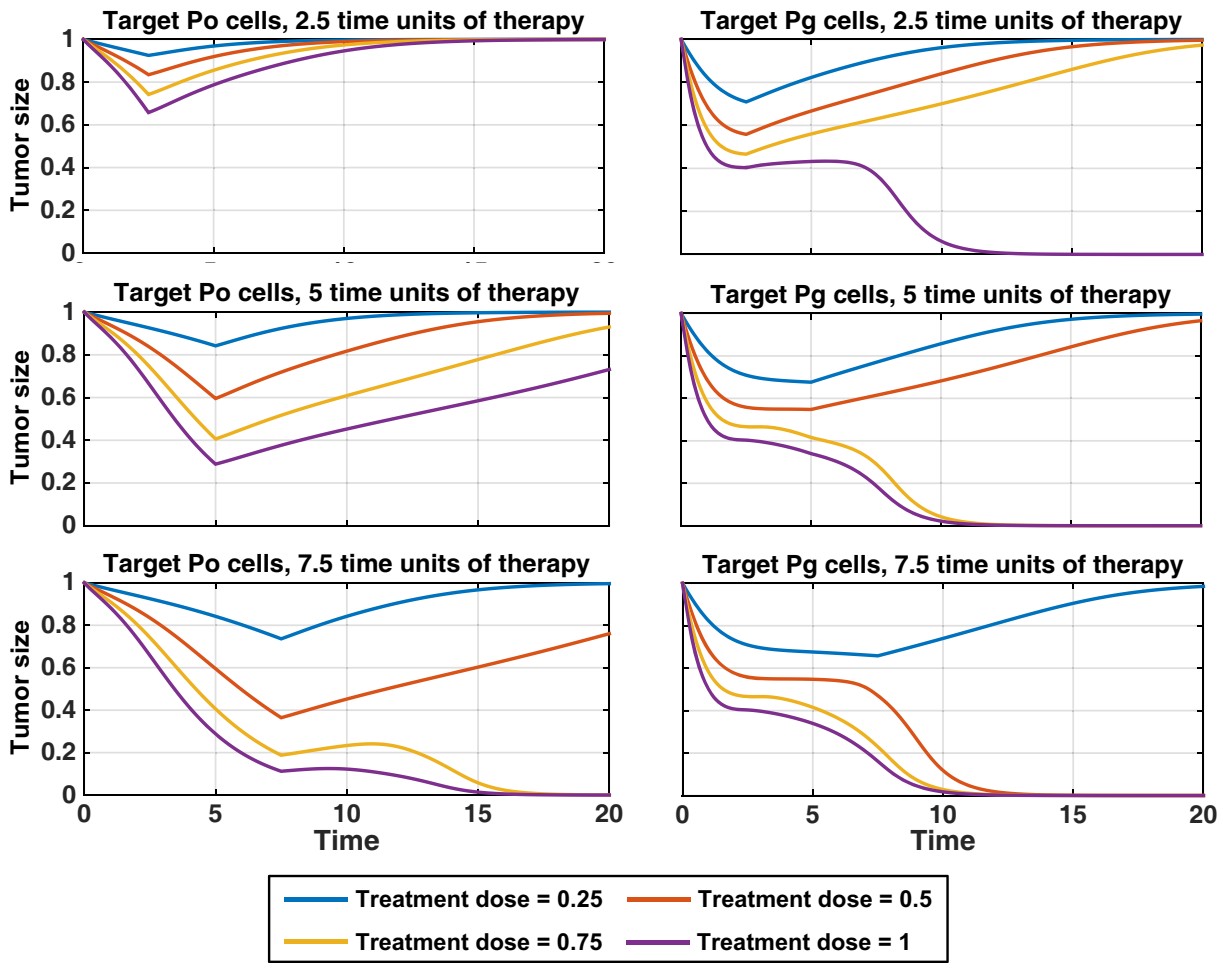

**Figure 5. Simulations identify the glycolytic cell population as a sensitive drug target.**
We target either cells with more oxidative phosphorylation ($P_o$; left) or cells with more glycolysis ($P_g$; right) selectively, starting from a metabolically patterned state, for 2.5, 5, or 7.5 (arbitrary) time units, with a treatment dose between 0.25 and 1. After therapy is halted, the cells are allowed to evolve according to the original model (Fig 2). The total cell populations, relative to the initial, starting cell population are shown. Corresponding populations of $P_o$ and $P_g$ cells can be found in Fig EV3.

Specifically, we targeted canonical Wnt signaling and PDK enzyme activity, both of which act as regulators of glycolysis (Fig 6A). Dichloroacetate (DCA) inhibits PDK activity and therefore targets cell metabolism directly by releasing inhibition of PDC, which increases OXPHOS capacity. Tankyrase inhibitors such as XAV939 reduce β-catenin levels and hence reduce canonical Wnt signaling. Compounds that target Wnt and PDK are currently being tested in preclinical studies as individual agents in clinical trials, but they have not been tested in combination (Fig 6B). We asked whether targeting glycolytic cells using anti-Wnt and anti-PDK therapies in combination is more effective than single-agent therapy.

Because the inhibition of β-catenin by XAV939 is similar to the effects of dnLEF-1 and dnTCF-1, we modeled treatment by XAV939 using an analogous approach. In particular, we assumed that XAV939 decreases the general Wnt signaling term $S_W$ and increases the ranges of Wnt and its inhibitor (due to upregulation of Wnt and Wnt inhibitor diffusers), which we modeled by increasing $D_W$ and $D_{WI}$ proportionally. To model the effects of DCA, we increased the rate at which cells switched from a glycolytic metabolic phenotype

to an OXPHOS phenotype (e.g., $1/\tau_{go}$ is increased) to reflect the tendency of cells to perform OXPHOS when PDK is inhibited.

In Fig 6C, we simulated a combination therapy applied to a fully developed tumor at steady state. At a fixed dose of XAV939, coupled to increasing doses of DCA, the simulation predicts that the population of oxidative cells will increase initially as cells switch from a glycolytic state ($P_g$) to an OXPHOS state ($P_o$) until there is an insufficient level of glycolytic cells to sustain the tumor and all the tumor cells die. Furthermore, the treatment simulations indicate that a combination of the two therapies will be more effective than single therapies as long as one or the other has adequately been applied. For example, a value of $1/\tau_{go} = 12$ is effective in eradicating the cells as long as Wnt signaling has been reduced by more than about 27% (e.g., $\bar{S}_W = 0.73$). In other words, if β-catenin expression has not been sufficiently suppressed by XAV939, then PDK inhibition by DCA must be adequately increased, and vice versa. Similar results are obtained when an *in vitro* version of the mathematical model is used to simulate the growth of colonies in fibrin gels (Fig 7C; see also Appendix A7 and Appendix Fig S17). In the *in vitro* case,

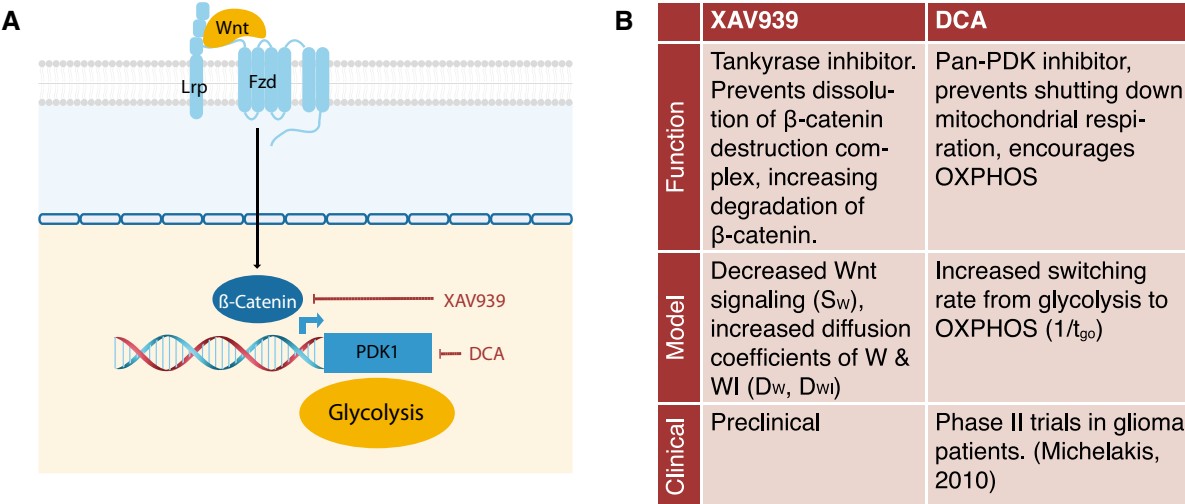

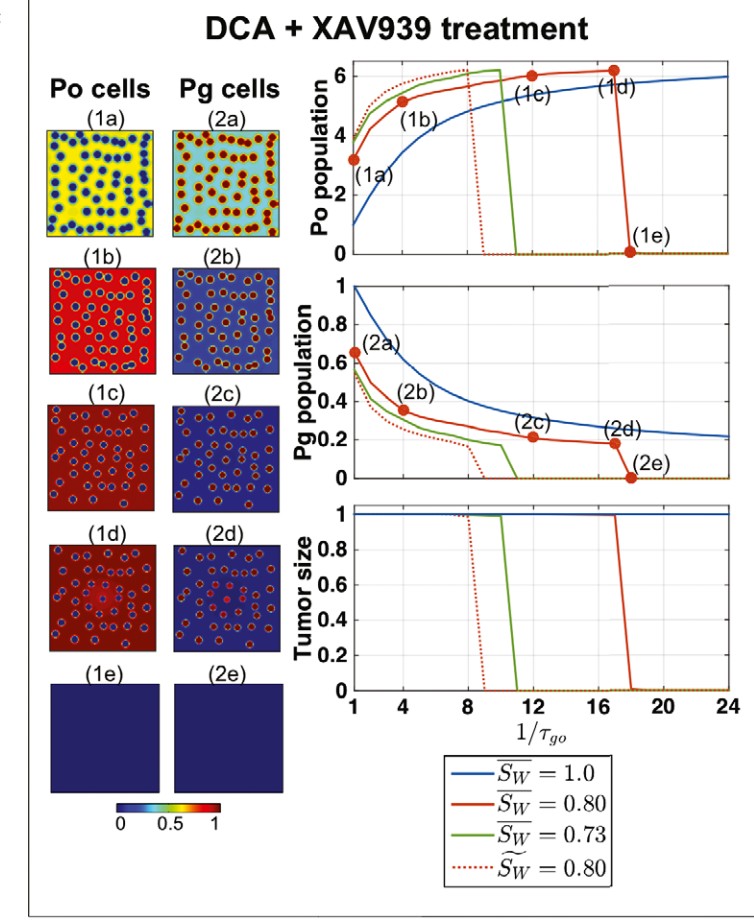

**Figure 6. Therapies targeting metabolism and Wnt synergize for tumor death in mathematical simulations.**

A, B  Modeled therapies, their targets, and the model parameters influenced by therapy.

C  Starting with a metabolically patterned state, treatment of tumors with dichloroacetic acid (DCA) and XAV939 combined leads to an effective crash in the system, as shown by the complete loss of cells (1e and 2e). The panels on the left show the cell arrangements for the oxidative ($P_o$) and glycolytic ($P_g$) populations (metabolic patterning), and the three graphs on the right show the fractions of $P_o$ or $P_g$ cells relative to their initial values, after applying the therapy for 50 (arbitrary) time units. The effects of therapy on the total cell population, relative to the initial cell population, for the same $D_W$ and $1/\tau_{go}$ values, are shown in the third graph. XAV939 treatment is modeled by decreasing $S_W$ and increasing $D_W$ and $D_{Wl}$ linearly with respect to the decrease in $S_W$; legend values are listed relative to mock $S_W$. The dashed curves labeled $\widetilde{S_W} = 0.80$ correspond to the case in which $\overline{S_W} = 0.80$, but $D_W$ and $D_{Wl}$ are unchanged and take the values used in the mock tumor simulations. The panels on the left correspond to the red curve in the graphs and show the effect on patterning for $1/\tau_{go}$ values 1, 4, 12, 17, and 18, respectively [denoted by labels (1a) through (2e)]. Color bar refers to unitless concentrations.

however, the colony does not die out as the DCA concentration is increased. Instead, the effect saturates because sufficient nutrients are available to diffuse throughout the spheroid to maintain cell viability in the absence of glycolytic cells. Additionally, in the *in vitro* version of the model, there is no angiogenesis, but there is cross-feeding between the OXPHOS and glycolytic cells (Appendix A7.1–A7.3).

Given that factors that increase Wnt diffusion were upregulated in the dnLEF/dnTCF tumors, we also tested the effect of increases in Wnt diffusion. We determined that increased expression of Wnt diffusers decreases the sensitivity of the tumors to treatment. For example, decreasing $S_W$ but maintaining $D_W$ and $D_{WI}$ at their pre-treatment values to model the inhibition of Wnt diffusers results in more efficient treatment—tumor eradication occurs at smaller concentrations of XAV939 and DCA (e.g., compare the red solid and dashed curves in Fig 6C (right panels), which correspond to a 20% reduction in Wnt signaling; the dashed curve shows the result when the production of Wnt diffusers is inhibited and is labeled as $\tilde{S}_W$).

We performed a preliminary experimental test of model predictions using 3D colony growth of colon cancer cells. A total of 200 single cells were seeded in fibrin gels and cultured under drug treatment for 14 days (Fig 7A). Over this time, the single cells proliferated to give rise to tumor spheroids, and we used image analysis to quantify the increase in colony size as a proxy for proliferation (Fig 7B and C). Treatment with low doses of DCA (0.5 mM, 2 mM) or XAV939 (5 μM) as single agents had no effect on the development and growth of colonies over the 2-week treatment period—in fact, DCA treatment appeared to increase colony size. By contrast, combination therapy had a significant inhibitory effect on colony growth, indicating a strong, negative, and synergistic effect on proliferation.

Synergy can be quantified using the Bliss Independence Model Combination Index (Foucquier & Guedj, 2015), which assumes that XAV939 and DCA treatments act independently (e.g., XAV939 targets Wnt signaling, and DCA targets PDK activity). In particular, if the Combination Index is less than one, this indicates synergy (see Appendix A8 for the definition of the Combination Index and further details). In the *in vivo* simulation and the *in vitro* experiments, the Combination Index is zero because neither XAV939 nor DCA treatment separately affects tumor sizes (provided the concentrations of XAV939 and DCA are not too large; Appendix A8). In the *in vitro* model, the Bliss Combination Index is 0.3462 (using $\bar{S}_W = 0.80$, $1/\tau_{go} = \frac{1}{4}$), although the Combination Index does depend on the drug concentrations and increases toward one as the DCA concentration increases because the responsiveness to DCA treatment saturates (Appendix A8). Since the Combination Indices in all cases are less than one, this indicates synergy of the XAV939 and DCA combinatorial treatments. As predicted by both the *in vivo* and *in vitro* models, these results suggest that combining Wnt inhibitors and metabolic targeting agents is a promising strategy for treating colon tumors.

## Discussion

In this study, we generated colon cancer xenograft tumors and examined changes in cellular metabolism by immunohistochemical staining for phospho-PDH (a marker of PDK activity), and markers of Wnt signaling (LEF-1, β-catenin). We observed that the tumors exhibit a pronounced spotted pattern of metabolic states where the spots indicate clusters of cells in which their mitochondria are inhibited (by PDK action) and thus where glycolysis was likely to be highly active. This is in contrast to the cells in the regions surrounding the spots. In these regions, mitochondria are more active (not inhibited by PDK) and therefore utilize more oxidative phosphorylation. Although we cannot rule out that the spotted pattern is due to the emergence of genetically distinct, clonal populations, the short timescale of the xenografting (14–21 days) and the reproducibility of the pattern in another cell line (SW620) as well as site of injection (subcutaneous and orthotopic within the colon cecum) suggest that what we have observed is a fundamental pattern of tumor heterogeneity that is not genetic in nature, but nongenetic and dynamic (Appendix A9 and Appendix Fig S20).

Metabolic patterning had been previously proposed as a mechanism to facilitate transport of glucose into hypoxic regions of tumors (Sonveaux *et al*, 2008). In particular, cells performing OXPHOS would be located near blood vessels, and, rather than fueling respiration using glucose, these OXPHOS cells would instead use lactate produced by the hypoxic (glycolytic) cells as an alternative nutrient source. Utilization of lactate frees the glucose to travel farther into the hypoxic regions of the tumor where it would be used during glycolysis. Recent studies propose that cancer cell subpopulations segregate and reorganize to survive the sudden loss of nutrients due to antiangiogenic therapy and that this leads to the development of resistance (Allen *et al*, 2016; Jiménez-Valerio *et al*, 2016; Pisarsky *et al*, 2016). However, in our xenograft images, we did not observe this type of spatial relationship between metabolism and vasculature. In fact, we observed that the spotted pattern was denser at the tumor margin where the vascular density was highest.

To investigate the mechanisms underlying the patterning we observed, we proposed a mathematical model based on Turing–Gierer–Meinhardt activator–inhibitor equations that simulated a symbiotic spatiotemporal relationship between these two cell populations (an oxidative population and a glycolytic population). Our model incorporated terms for Wnt as an activator with a short range of diffusion and Wnt inhibitors (e.g., SFRP, DKK) with longer ranges of diffusion. The Wnt and Wnt inhibitor equations describe a feedback relationship, and this lies at the crux of a spotted pattern that emerges in our simulations. Our equations also describe activities of metabolic reprogramming through changes in Wnt levels and availability of nutrients and cell substrates. The model describes a mutualistic interaction between the glycolytic and OXPHOS cells because the glycolytic cells induce the delivery of nutrients from blood vessels to mimic the effects of lactate-induced angiogenesis and these nutrients benefit both cell types. More generally, we can also interpret these nutrients as mutually beneficial cell substrates produced by the glycolytic cells. When we considered the effects of symbiosis by explicitly incorporating cross-feeding between glycolytic and OXPHOS cells in a more general model (Appendix A3) and an *in vitro* model (Appendix A7), we found that Wnt signaling dominates the behavior and the patterning is robust to this form of symbiosis. Although mathematical models have been developed previously to investigate metabolic symbiosis, including spatially homogeneous (Mendoza-Juez *et al*, 2012) and heterogeneous models (McGillen *et al*, 2014; Phipps *et al*, 2015), to our knowledge,

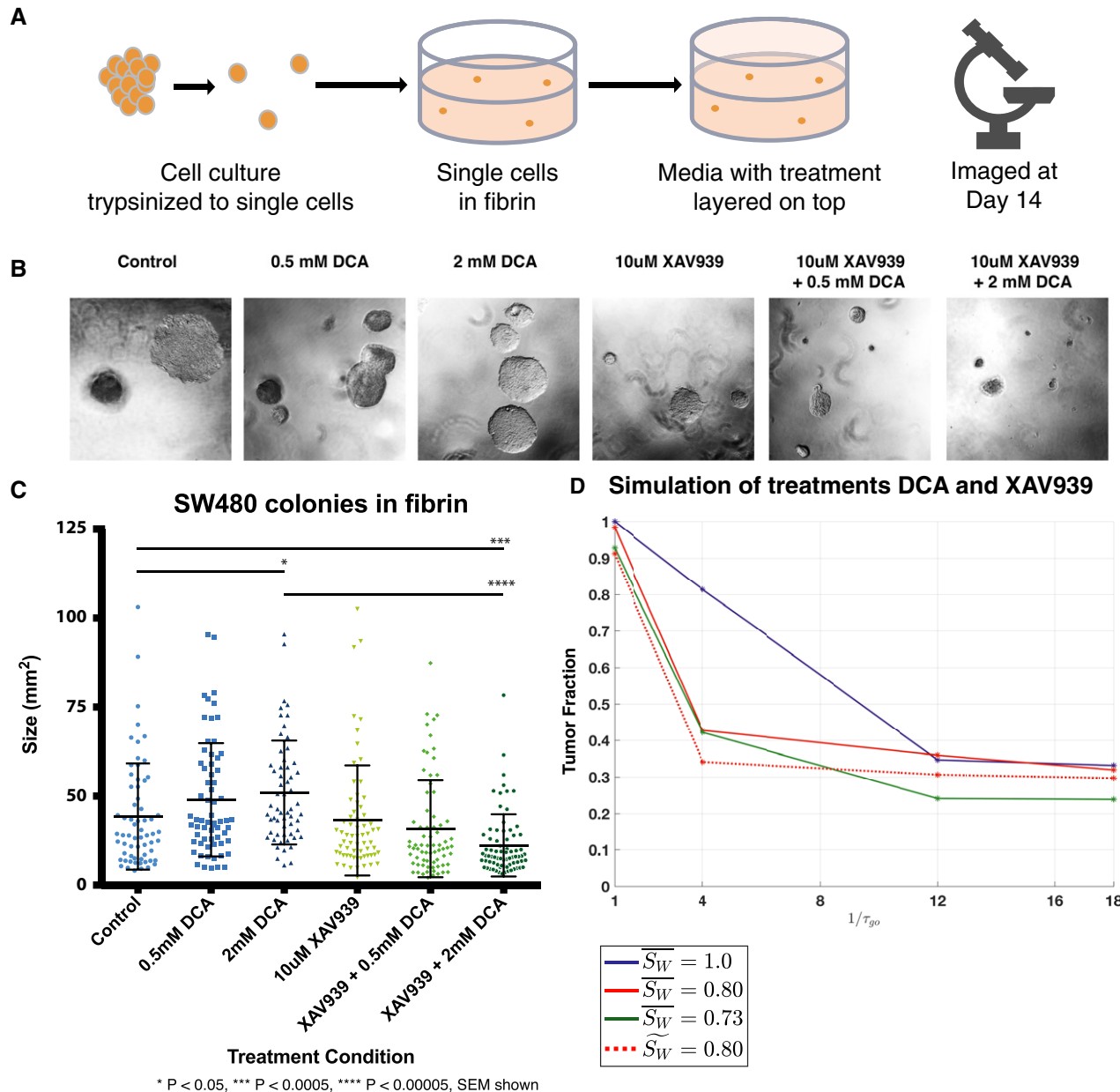

**Figure 7.  Targeted therapy significantly decreases SW480 tumor spheroid size *in vitro*.**
Combined Wnt signaling and glycolysis targeting therapies significantly decrease SW480 spheroid size *in vitro*.

A    SW480 cells were embedded in a fibrin gel using the method shown. Media containing a mock treatment, 0.5 mM DCA, 2 mM DCA, 10 μM of XAV939, or a combination of DCA and XAV939.

B    Representative 4× images of spheroids each condition, imaged 14 days after treatment, are shown.

C    Analysis of 75 spheroids per condition shows 2 mM DCA significantly increases SW480 spheroid size, while combined 2 mM DCA treatment with 10 μM XAV939 significantly decreases their size. Statistical significance was determined using Student's two-tailed *t*-test.

D    The effects of therapy on the total cell population, relative to the initial cell population, of combined XAV939 and DCA treatment were simulated using an *in vitro* version of the model (Appendix A7). As in Fig 6, DCA treatment was modeled by increasing the rate at which cells switched from a glycolytic metabolic phenotype to an OXPHOS phenotype (e.g., $1/\tau_{go}$ is increased) to reflect the tendency of cells to perform OXPHOS when PDK is inhibited. XAV939 treatment was modeled by decreasing the general Wnt signaling term $S_W$ and increasing the range of Wnt and its inhibitor (due to upregulation of Wnt and Wnt inhibitor diffusers), which we modeled by increasing $D_W$ and $D_{WI}$ proportionally. The dashed curves labeled $\tilde{S}_W = 0.80$ correspond to the case in which $\bar{S}_W = 0.80$ but $D_W$ and $D_{WI}$ are unchanged.

the model presented here is the first to describe a pattern for heterogeneity in tumors that derives from an intricate spatial relationship between metabolic types and Wnt signaling. Though the role of Wnt signaling in cancer growth and development has been studied

for many years, only recently has its regulation of Warburg/glycolysis metabolism been described. A better understanding of the link between Wnt and metabolism is crucial for defining how this overactive pathway drives tumorigenesis and progression, and for

developing novel cancer treatments that target its key oncogenic actions.

Our mathematical model demonstrated strong qualitative and quantitative (spatial) agreement with the spotted patterns of activity detected in the tumors. The model also predicted that interference with Wnt signaling is not solely the result of a decrease in overall Wnt activity. Making the simple parameter change of decreasing Wnt signaling throughput leads to overall less glycolytic activity (lower background of $P_g$ cells), a prediction that was validated in our xenograft experiments in which Wnt transcription was partially blocked by the overexpression of dominant negative LEF-1. However, the phospho-PDH stains also revealed fewer, but larger regions of PDK activity (i.e., larger, fewer cell clusters). To simulate these observations, coefficients of diffusion for Wnt and its secreted inhibitors were increased—parameter changes that resulted in a "spreading out" of Wnt and Wnt inhibitor activity. This model prediction prompted us to investigate whether the expression of proteins known to increase the range of Wnt ligand and Wnt inhibitor diffusion was increased when dnLEF- or dnTCF-expressing colon cancer cells were developed into xenograft tumors. Consistent with the mathematical model, we observed that the diffusers SPOCK2 and GPC4 were overexpressed in our xenografts but interestingly, not in our 2D *in vitro* culture conditions. The expression of SFRP5, which acts simultaneously as a Wnt inhibitor and diffuser by preventing binding to Frizzled receptors, also shows somewhat higher expression in our Wnt-low xenografts (dnLEF-1). It is important to emphasize that it is human-specific oligonucleotide primers that detect these changes in expression of Wnt ligand modifiers. Thus, the implanted human colon cancer cells appear to adapt to interference with Wnt signaling by directly increasing expression of Wnt ligand regulators. Our analysis of primary human colorectal tumors stressed by radiochemotherapy treatment shows that one consequence of therapy could be a similar increase in the distribution of Wnt ligands through upregulated expression of glypicans (e.g., GPC1) and SFRP proteins (Fig 4E). These observations suggest that there are likely to be significant changes in Wnt signaling dynamics and metabolic programming of treated tumors, perhaps as a means of coping and surviving the loss of nutrient and cellular damage.

The model also suggested that the tumor is most reliant on the glycolytic cells, and we found that inhibitors that target glycolysis and Wnt signaling in combination are more effective than treatments that only target one of these features. We simulated the actions of XAV939, which lowers β-catenin levels, and DCA, which inhibits PDK activity, thereby decreasing glycolysis. Since standard-of-care treatments for colon cancer have not changed significantly for decades, the novel combination of an inhibitor of glycolysis (e.g., DCA) with a Wnt pathway inhibitor (e.g., XAV939) might be an effective treatment to consider. We validated this prediction using 3D colony growth of SW480 colon cancer cells, which have high, intrinsic Wnt signaling and high levels of glycolytic activity. However, given that primary human colon cancers are more complex with respect to intrinsic Wnt pathway activity and crosstalk with the microenvironment, determining which tumor subtypes will be the most sensitive to this drug combination and how *in vivo* tumors respond to treatment will be important issues to resolve in developing effective drug combinations.

Our mathematical model is an abstract idealization and simplification of tumor proliferation and metabolism, and real tumors are much more complex than modeled here. For example, it is likely that Wnt and a Wnt inhibitor are not the only factors contributing to the pattern, although it is clear that Wnt has a strong influence, since the spots change significantly when Wnt signaling is interfered with. In the Appendix, we developed a more complete model that simulated PDK activity, hypoxia-inducible transcription factor concentrations (HIF1α), and lactate concentration (see Appendix A3), and linked these factors to cross-feeding, Wnt signaling, and metabolism. In this more detailed model, the switch between metabolic phenotypes depends on PDK activity, rather than on Wnt directly. However, since Wnt and HIF1α promote PDK expression and activity (Kim *et al*, 2006; Pate *et al*, 2014; Prigione *et al*, 2014), we found that the spatiotemporal distributions of PDK, Wnt, and lactate track closely together and so the results from the more detailed model are qualitatively similar to those presented in the main text for the simplified model.

While we have used a Turing-type model to simulate the spotted pattern, a decision well supported by data, we acknowledge that this type of model is only one possible mechanism to explain the patterning in the tumors. Other alternatives include differential adhesion or cell sorting—a process where cells of different adhesion potential sort away from each other (Amack & Manning, 2012; Foty & Steinberg, 2013), and bet-hedging—a process where different cells are differentially sensitive to external stresses, so that no matter the condition, at least some cells thrive (de Jong *et al*, 2011; Starrfelt & Kokko, 2012; Vogt, 2015). For bet-hedging to generate patterns, cell state changes must be reversible on a time scale slower than cell division, so that cells of like state end up clustered by default. Further experiments are needed to definitively distinguish among these processes. For example, these three types of processes (Turing, differential adhesion, bet-hedging) would be expected to be driven by different types of molecular signals and so these signals would need to be identified and tested.

Finally, although we focused on xenograft tumors, artificial constructs that only partially model tumorigenesis, we observed heterogeneity in metabolism and Wnt signaling in other settings as well. For example, we generated orthotopic tumors by implanting colon cancer cells (mock-parental SW480 and SW620, as well as dnLEF- or dnTCF-expressing variants) in the wall of the mouse cecum and observed patterning (Appendix A9 and Appendix Fig S19). In normal colon epithelia, pPDH stains show a gradient with high PDK activity in the base of the crypt where there is strong Wnt signaling and less PDK activity at the top of the crypt near the mucosal surface where Wnt signaling is not active (Pate *et al*, 2014; Fig 1D). In primary human patient colon tumors, our analysis revealed clear and striking heterogeneity in PDK activity and LEF-1 expression in the epithelial portion of the tumor (Fig 1D). While this heterogeneity was not apparent as a regular array of cell clusters like the xenograft patterns, groups of cells with markedly different activities are clearly evident. Since the tumor microenvironment is more complex in human tumors than xenograft tumors, an inherent pattern of metabolic and Wnt activity may be modified by additional structural and cellular components as well as nutritional stresses and a changing microenvironment. Understanding how these additional components influence metabolic heterogeneity and the symbiosis between neighboring cells and how they might create more complex pattern-on-pattern activities is a challenge going forward.

# Materials and Methods

### Numerical simulations

The nondimensionalized equations for the rate of change in the population of oxidative ($P_o$) cells and glycolytic ($P_g$) cells, respectively, are shown in Fig 2A. See Appendix A2 for details on the nondimensionalization. Figure 2B shows the rate of change in the concentration of Wnt and Wnt inhibitor (W and $W_I$) activity, respectively.

The first term on the right side of the equality in the $P_o$ and $P_g$ equations refers to diffusion, or random motion of the cells. The next terms are standard logistic proliferation terms, with proliferation dependent on nutrient level N, cell type, and the current total population of cells in the domain. The model sets the proliferation of glycolytic cells to be dependent on Wnt activity according to Michaelis–Menten dynamics, given by the term $W/(\alpha_W+W)$, which saturates at high levels of W. The parameters $1/\tau_o$ and $1/\tau_g$ are proliferation rates and $1/\tau_{go}$ and $1/\tau_{og}$ are switching rates. The last terms in these equations are cell death terms; death is modeled such that it occurs if the nutrient supply N drops below a threshold $N_d$. The death rates are given by $\mu_o$ and $\mu_g$.

The model is designed such that glycolytic and oxidative cells can emphasize, or "switch", metabolism programs depending on W, Wnt activity, which is reflected in the third and fourth terms of each equation, where $\chi_W$ and $\chi_W^*$ are switch functions. Each switch function is defined by a modified hyperbolic tangent function, such that if Wnt activity falls below a parameter W*, then the cells utilize a more dominant OXPHOS program, and if Wnt activity is above W*, then cells are more likely to utilize a greater level of glycolysis.

We assume that oxidative cells can switch to utilizing more glycolysis only if sufficient nutrient is present, given by parameter $N_g^*$. Cells will die if nutrient is below the parameter $N_d$. The steepness of the functions can be adjusted so that they are more step-like and hence more sensitive to W and N. Since we use large values for the steepness of the functions, we could alternatively have used piecewise functions for $\chi_N$ and $\chi_N^*$.

The dynamics of dead cells (not shown) is described by a similar reaction–diffusion equation. This is the population of cells that have died from lack of nutrient. These cells can also diffuse and decay. The equations in Fig 2 describe W (Wnt) and $W_I$ (Wnt inhibitor, e.g., DKK or SFRP) activity. $D_W$ and $D_{WI}$ are constant diffusion coefficients. It has been shown in epidermal cells that Wnt target genes produce Wnt signals as well as long-range secreted Wnt inhibitors (Lim *et al*, 2013), so the inhibitor is assumed to diffuse much longer range than Wnt; that is, $D_W$ must be significantly smaller than $D_{WI}$. Wnt signaling activity is assumed to be nonlinear with respect to Wnt and is inhibited by the Wnt inhibitor through the term $1/(a+b W_I)$. We assume the Wnt inhibitor is being produced by Wnt activity through both cell types. The terms $\mu_W$ and $\mu_{WI}$ are decay rates. The term $S_W(P_o+P_g)$ in the Wnt equation refers to constitutive Wnt signaling through the cells.

The equation for nutrient (eq. 1h in Appendix A2) describes the diffusion and uptake, decay, and source of nutrient. The nutrient term has Dirichlet (fixed) boundary conditions and diffuses in from the boundary of the spatial domain, so that the boundary can be considered as regions where vasculature is high. The second and third terms refer to uptake of nutrient by the two different cell types. The term $\mu_N N$ is a natural decay term. The last term, $N_S$, refers to the nutrient source, which is a small source term applied to the entire domain. This source term is based linearly on the glycolytic activity of the cells and is given by $N_S = N_S(\int P_g) = \gamma_N[(1-\alpha_N) \int \frac{P_g}{S_x S_y} + \alpha_N]$, where $\alpha_N$ and $\gamma_N$ are parameters, $\int P_g$ is the integral of $P_g$ cells, and $S_x$ and $S_y$ are the lengths of the sides of the spatial domain. This function was chosen so that $N_s(0) = \gamma_N^{\alpha N}$ and $N_s(S_x S_y) = \gamma_N$ ($S_x S_y$ is the maximum that $\int P_g$ can reach). We chose to have the nutrient source $N_s$ depend on $P_g$ cells because glycolysis induces angiogenesis (Dhup *et al*, 2012; Porporato *et al*, 2012; Ruan & Kazlauskas, 2013), allowing more nutrients and growth factors to be delivered to the tumor (Pate *et al*, 2014). We use a linear function in the model as the simplest form for the dependency between N and $P_g$, which is consistent with experimental observations.

To summarize, in addition to consideration of Wnt signaling dynamics, biological assumptions for the model include terms for random motion in space (diffusion), terms for each cell type (oxidative and glycolytic, or $P_o$ and $P_g$, respectively), and their propensity to proliferate, die, and switch to the other cell type. Equations were included to account for dead cells, which consists of $P_o$ and $P_g$ cells that have died from lack of nutrient, and which can diffuse and decay. Terms for Wnt (W) and Wnt inhibitor ($W_I$) activity were made nonlinear with respect to Wnt, meaning that their rates are proportional to Wnt activity. Nonlinear Wnt activity is dependent on $P_g$ levels, while nonlinear Wnt inhibitor activity is proportional to both $P_g$ and $P_o$ levels. A term for constitutive Wnt signaling was included for both cell types as well as decay terms for W and $W_I$. The general nutrient term N can diffuse, decay, and be taken up by the different cell populations. A bulk source was included for the nutrient as well as a Dirichlet boundary condition, both of which are dependent on the average level of glycolytic cells in the domain, a simplified way to incorporate increased angiogenesis driven by glycolysis (Pate *et al*, 2014). This relationship was included to take into account our observation that there is considerably less vasculature in tumors in which Wnt signaling has been blocked by dominant interfering forms of the Wnt transcription factors lymphoid enhancer factor-1 (dnLEF-1) or T cell factor 1 (dnTCF-1) (Pate *et al*, 2014). Finally, there is a baseline assumption that there is sufficient oxygen available throughout the domain for OXPHOS to operate, even at a minimal level.

In the numerical results presented here, no-flux boundary conditions were used for all terms except N, which is governed by Dirichlet boundary conditions (N at the boundary is equal to the value $\frac{1}{\gamma_N} N_s$ where $N_s$ is described above). Initial conditions were set for a random distribution of $P_g$ cells located near the boundary, and small random values of W and $W_I$ in the same areas where initial $P_g$ cells are located. A constant high level of nutrient throughout the domain was provided (results did not change qualitatively if N was solved as a quasi-steady-state equation), and the initial condition contained no $P_o$ or $P_d$ cells. All parameters are given in Table 1, and a sensitivity analysis is provided in Appendix A5.

Numerical simulations were performed in MATLAB, using a forward difference method for each time derivative. $P_o$, $P_g$, W, and $W_I$ equations were solved implicitly in centered diffusion terms. The nutrient equation was solved implicitly in uptake, decay, and centered diffusion terms.

## Animal protocols for xenograft and orthotopic tumors

SW480 stable transductants for xenograft or orthotopic injection were prepared through lentiviral infection with pCDH vector from System Biosciences: empty vector (mock), or vector expressing dnLEF-1 or dnTCF-1, followed by selection with 500 μg/ml G418. Transduced cells were collected as a pool for confirmation of expression, and Wnt signaling activity was measured by a SuperTOPFlash luciferase reporter (Pate *et al*, 2014). A total of $2.5 \times 10^6$ cells were injected into immunodeficient NSG mice [2-month-old NSG male and female mice were used for the subcutaneous xenograft tumors (JAX™ Mice from Jackson Labs); male and female NSG mice, approximately 3 months old, were used for orthotopic tumors (injection of $5–10 \times 10^3$ cells into the cecum wall)]. Tumors were removed (subcutaneous after 3 weeks, orthotopic after 4 weeks), fixed in paraformaldehyde overnight, and paraffin-embedded 4 weeks after injection. All experiments involving animals were approved by the UCI IACUC (Protocol 2002-2357-4 to R. Edwards).

## Immunohistochemistry

Deparaffinized 5- to 6-μm sections of formalin-fixed paraffin-embedded (FFPE) mouse xenograft tumor and human colorectal carcinoma tissues followed by pressure cooker antigen retrieval in citrate buffer were blocked in 3% $H_2O_2$ and goat or horse serum plus MOM block reagent (if mouse primary antibody was used on mouse tissue), avidin, and biotin blocking reagents (Vector Labs). Sections were incubated in primary antibodies: antiphospho-PDHpSer293 (Calbiochem; 1:50–1:100), anti-β-catenin (BD; 1:500), anti-LEF-1 (Cell Signaling; 1:100), anti-HIF1α (Thermo Scientific; 1:1,000) followed by biotinylated secondary antibodies and visualization using a peroxidase-conjugated avidin-based Vectastain protocol. Slides were then counterstained with hematoxylin and mounted using Permount mounting medium (Fisher). Images were captured using an Olympus FSX100 system and processed in Adobe Photoshop.

## Quantitative PCR

RNA was extracted from xenograft tumors and cells using TRIzol (Invitrogen) following the manufacturer's instructions. cDNA was synthesized with 1 μg of total RNA with the High Capacity cDNA Reverse Transcription Kit (Invitrogen), as per the manufacturer's instructions. qPCR was performed in triplicate for each experimental condition using Maxima SYBR Green qPCR Master Mix (Invitrogen), according to the manufacturer's instructions. To normalize mRNA levels, GAPDH probes were used. Primer pairs are as follows: GAPDH forward: TCGACAGTCAGCCGCATCTTCTT, reverse: GCG CCCAATACGACCAAATCC; TINAGL1 forward: ACCAGGTCACTC CTGTCTACC, reverse: GATGCCTCCCTTGTATAGGAAG; CDC42 forward: CCATCGGAATATGTACCGAC, reverse: CTCAGCGGTCG TAATCTGTC; SPOCK2 forward: CCCGGCAATTTCATGGAGG, reverse: GCGGTTCCAGTGCTTGATC; GPC1 forward: GGCTGGTGGCT GCTATGT, reverse: CAGGTTCTCCTCCATCTCGC; GPC2 forward: CACCTGCTGTTCCAGTGAGA, reverse: AGAGAGTGCTGGGCTACT GA; GPC4 forward: GTGGGAAATGTGAACCTGGAA, reverse: CGAG GGACATCTCCGAAGG; DKK4 forward: GGGACACTCTGTGTGAA

CGA, reverse: TGGTTTTCCTGGACTGGGTG; SFRP5 forward: CTGT ACGCGTCATCCTAGCC, reverse: CGGACCAGAAGGGGGTCTAT.

## Fibrin gel assay

A total of 200 trypsinized SW480 cells were mixed with 100 μl of 2.5 mg/ml bovine fibrinogen (MP Biomedicals) in DMEM plus 10% FBS and 1% penicillin–streptomycin–glutamine and 1 μl of thrombin (Sigma). The fibrin gels were seeded in 96-well, flat-bottom plates. After the gels solidified, 100 μl of DMEM media containing the desired drug treatment was layered on top (DCA was obtained from Sigma, XAV939 from Stemgent). Wells were imaged after 14 days of incubation. Size measurements were taken using Adobe Photoshop. Data were analyzed using Prism (GraphPad).

## Image processing of spots

Image processing (overlay of spot contours and convex hulls) was done using built-in functions in MATLAB's Image Processing Toolbox. Briefly, a color channel of an image was converted to a binary image based on a manually chosen threshold dependent on staining intensity. A noise filter was applied to reduce background staining. Thresholds were then chosen to define cutoff values of spot boundaries. Parameters for built-in tools were chosen manually to give the best fit for pattern contours. Details for this method are provided in Appendix A1 and A1.1–A1.3, and Appendix Figs S1–S10.

**Expanded View** for this article is available online.

## Acknowledgements

The authors would like to thank the Waterman, Donovan, and Lowengrub laboratories for their feedback and discussion. We would also like to thank Eric Stanbridge, Peter Donovan, Arthur Lander, Jun Allard, Olivier Cinquin, Jenny Wu, Harry Mangalam, Babak Shababa, and Charless Fowlkes for providing advice and critiques. The work of G.T.C. and M.L.W. was supported by NIH Grants CA096878, CA108697, CA200298, a California CRCC award CRR-17-429379, and a P30CA062203 to the Chao Family Comprehensive Cancer Center. The work of M.L., E.J.P., and J.S.L. was supported in part by NSF Grant DMS-1263796, NIH Grant P50-GM76516 for a Center of Excellence in Systems Biology at the University of California, Irvine, as well as the NIH Grant P30-CA062203 grant. In addition, the authors also received partial support from the University of California, Irvine, through a seed grant. This work was made possible, in part, through access to the Tissue Specimen Shared Resource and Genomics and High Throughput Facility of the Cancer Center Support Grant (CA-62203) at the University of California, Irvine, and NIH shared instrumentation grants 1S10RR025496-01 and 1S10OD010794-01. K.W. and R.A.E. were supported by P30CA062203.

## Author contributions

ML and JL conceived the mathematical model; ML and EP coded and ran all simulations; ML, GTC, JL, and MLW wrote the manuscript; MLW and GTC designed the biological experiments; GTC, KW, and RAE performed biological experiments.

## Conflict of interest

The authors declare that they have no conflict of interest.

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
