## [Review Process File · Molecular Systems Biology]

Mathematical modeling links Wnt signaling to emergent patterns of metabolism in colon cancer

Mary Lee, George T. Chen, Eric Puttock, Kehui Wang, Robert A. Edwards, Marian L. Waterman and John Lowengrub

Corresponding authors: John Lowengrub and Marian L. Waterman, University of California Irvine

Review timeline:

Submission date:	17 October 2016
Editorial Decision:	03 November 2016
Revision received:	16 December 2016
Editorial Decision:	22 December 2016
Revision received:	05 January 2017
Accepted:	12 January 2017

Editor: Maria Polychronidou

Transaction Report:

1st Editorial Decision

03 November 2016

Thank you again for submitting your work to Molecular Systems Biology. We have now heard back from the two referees who agreed to evaluate your study. Overall, the reviewers appreciate the interesting findings presented in the study. However, they raise a number of concerns, which we would ask you to address in a revision. The reviewers' recommendations are rather clear and I think that there is no need to repeat the points listed below. Please let me know in case you would like to further discuss any specific point.

REFeree REPORTS

Reviewer #1:

Lee et al. uncover an intriguing spatial pattern of spotted patches of PDK activity and Wnt signaling in tumor xenografts. They develop a Meinhardt Gierer-type mathematical model of pattern formation that explains this pattern through the dependency of Wnt signaling, cellular metabolic state and the secretion of Wnt inhibitors. They also demonstrate that perturbations to Wnt signaling change the spatial features of the patterns observed. The paper combines experiments in an in-vivo context with high level mathematical modelling and provides stimulating insights into tumor spatial heterogeneity. While the repeating spotted pattern observed in xenografts is not recapitulated in tumors, the phenomenon is interesting and may expose unappreciated modes of tumor cell metabolic symbiosis. I therefore believe the paper will be of interest to cancer biologists as well as to systems biologists studying tumor heterogeneity.

A major point that must be addressed is the experimental system for establishing a relation between

glycolysis and Wnt signaling. The premise of the paper and the basis for the modelling approach is that there is a correlation between the cellular glycolytic activity and Wnt signaling. The authors perform immunostaining of pPDH and of LEF1 as markers for glycolysis and Wnt signaling respectively. They then identify similar spatial statistics of the sizes and inter-patch distances for both of these markers and deduce correlation from this. This is not convincing, especially given the large variability in patch spatial statistics observed in Figure 1C. One could just as well obtain similar spatial statistics with a salt and pepper type of pattern (where the glycolytic cells are actually adjacent to the Wnt positive cells). What the authors should instead do is to perform immunostaining of pPDH and LEF1 on serial sections to demonstrate that the patches actually overlap.

Minor comments:

- Figure 1A should demonstrate the core-periphery gradients, e.g. by adding a zoomed in version of the periphery as well or some quantification of this feature, which is currently missing.

- Figure 1B is unclear (especially the zoomed in versions on the right). The image contains blue and red contours which are not explained (is this a result of automated detection)? At the minimum these contours should be added to the higher magnification panels.

- Figure 1D is also unclear, the authors should show a zoomed in version demonstrating areas of positive and negative staining. Some quantification is also needed to demonstrate that there is spatial heterogeneity in expression levels (how would staining look like in a regular tissue or a cell line). Without some positive or negative controls the panel shown does not demonstrate heterogeneity and it would actually be better to remove this entire claim.

- The text is missing a reference to Figure 1D when discussing the tumor staining.

- Figure 1 - the authors should quantify the size of the patches not only in microns but also in number of cells per patch.

- Table of parameters in Figure 2 - The references are missing from the paper.

- Model equations in Figure 2 - define N and the chi function.

- Figure 4D,E - is there any change in the expression of glycolysis genes?

- When presenting the mathematical model in the main text the authors should define W_i (the Wnt inhibitor) and S_w (the rate of Wnt production) .

- In the Discussion the authors state that the cells are genetically identical. Tumors cannot be assumed to be genetically identical due to clonal heterogeneity and indeed the authors later discuss alternative mechanisms, e.g. cell sorting that can be genetic in nature, the authors should explain this claim of non-genetic effects, e.g. by assuming a genetic bottleneck in the xenograft model.

- The authors should elaborate on the method of spot detection in the Methods section.

- The caption of Figure 3 contains several typos.

- The choice of 161 μ m for a scale bar in the images is quite unconventional.

- Figure caption 6 mentioned B.1e and B.2e which I could not find.

Reviewer #2:

In "Mathematical modeling links Wnt signaling to emergent patterns of metabolism in colon cancer", Lee et al. computationally modeled a glycolytic "spotting" pattern that arises in colon cancer xenografts using a Turing-type reaction diffusion model mediated by Wnt and its inhibitor. Factors including angiogenesis, cell turnover and nutrient availability were also considered.

Simulation of this model predicted an intratumoral metabolic heterogeneity in which glycolytic and oxidative phosphorylation cells coexist and form a regular spatial structure, which was also consistent IHC in both xenograft tumor models and primary human tumors. Overall, this study was rigorously conducted and advances our understanding of interplay between signal transduction and metabolism in cancer by extending this known concept which is mostly studied in cell autonomous contexts to investigate spatial structure in the multicellular tumor environment - a subject of intense interest. The findings are novel and will be of broad interest to systems biologists and cancer biologists. Some considerations should however be addressed.

1. Cross-feeding (i.e. the production of lactate in one cell and its use as an oxidative carbon source in another) is a well-known phenomenon in metabolic symbiosis, as the authors cited but it was not considered in the model. Will cross-feeding between glycolytic and OXPHOS cells alter the resulting spatial distribution of these cells? This could be addressed by adding a new type of nutrient produced by P_G that feeds P_O. The authors state in the Discussion that "we can also interpret these nutrients as mutually beneficial cell substrates produced by the glycolytic cells", but this should be directly evaluated. Nutrients produced by glycolytic cells are unlikely to contribute to proliferation of the glycolytic cells, but the model didn't distinguish between nutrients.

2. The Wnt inhibitor/glycolysis inhibitor combination appears to be synergistic in a 3D in vitro system. However, the coexistence of glycolytic and OXPHOS cells in a regular spatial structure, as well as response to interference of Wnt signaling was validated by in vivo models. The 3D culture system may not be an appropriate system to test this model, since it lacks an important factor included in the mathematical model, that is, angiogenesis triggered by lactate produced in glycolytic cells, which increases nutrient availability of neighboring cells. I'm not sure if this factor is critical for the model to predict synergy between Wnt and glycolysis inhibition. It will be better if the authors could validate the synergy in in vivo tumor models or show that the synergy is robust to whether this term is included.

Minor:

Given the importance of SW as a surrogate of Wnt signaling activity, it is important to explain this parameter earlier in the text (it is currently introduced on 7th page)

Figures 1-3: It would be useful to score the IHC staining just to show the level of heterogeneity in the patterning, especially post lentiviral transduction.

Figure 3E: It would be more appropriate to average the data points for subsequent comparison with the simulation.

Figure 6C: What happens to combination response when the Wnt signaling is reduced by more than 30%? 50%? Etc. More rigorous metrics to assess synergy between the treatments would improve this analysis (e.g. the Bliss score or another one). Also, the right panel of Figure 6C is not explained clearly. What is the difference between $(S_W)^+$ and $(S_W)^-$, which seems to be the only difference between the red dashed lines and the red solid lines?

It is suggested to include a figure showing the experimental results on combinational inhibition of Wnt and PDK1 in the manuscript instead of putting the related figures in Supplementary Materials which makes these results difficult to follow.

Appendix Figure S25 & S28: should clarify what statistical tests were used.

Continued on next page.

We thank the reviewers for a positive, constructive review of our study on metabolism and tumor heterogeneity; they have provided valuable critiques and excellent suggestions. Overall reviewers felt that the discovery of a spotted, heterogeneous pattern of metabolism and Wnt signaling is a novel finding and that our work to mathematically model this pattern in a xenograft setting will be of interest to system biologists and cancer biologists. However reviewers had three major issues as well as minor comments. Below we provide a response to all issues and comments by providing new data. Changes to the manuscript text are indicated in blue font.

Major Comments

1. Reviewer 1:

A major point that must be addressed is the experimental system for establishing a relation between glycolysis and Wnt signaling. The premise of the paper and the basis for the modelling approach is that there is a correlation between the cellular glycolytic activity and Wnt signaling. The authors perform immunostaining of pPDH and of LEF1 as markers for glycolysis and Wnt signaling respectively. They then identify similar spatial statistics of the sizes and inter-patch distances for both of these markers and deduce correlation from this. This is not convincing, especially given the large variability in patch spatial statistics observed in Figure 1C. One could just as well obtain similar spatial statistics with a salt and pepper type of pattern (where the glycolytic cells are actually adjacent to the Wnt positive cells). What the authors should instead do is to perform immunostaining of pPDH and LEF1 on serial sections to demonstrate that the patches actually overlap.

To address the reviewer's concern, we have carried out additional experiments. First, we have repeated the staining for phospho-PDH and LEF1 on serial sections of Mock xenograft tumors (SW480 parental cells transduced with empty lentivirus; Figure 1A, 1B). As before we provide progressively higher power zoom-in panels to show the clusters of cells (spots), as well as a sample of the automated image analysis used to calculate size and distance to nearest neighbor.

In a new figure (Expanded View, EV1), we show the contour analyses and an overlay of the serial sections, which we then use to calculate the percentage of overlap. Overlap will never be 100% due to the use of serial sections and therefore imperfect alignment. Nevertheless, we observe a nearly constant percentage of overlap (65%-75%) whether we use low or high thresholds for identifying cell clusters, or spots. This indicates that the overlap is significant and not due to a random salt-and-pepper type of pattern.

In addition, we provide data from a new experiment to detect phospho-PDH and LEF1 in serial sections of primary human colon cancer (Figure 1E). The results show striking co-localization of positivity for both antigens.

2. Reviewer 2:

1. *Cross-feeding (i.e. the production of lactate in one cell and its use as an oxidative carbon source in another) is a well-known phenomenon in metabolic symbiosis, as the authors cited but it was not considered in the model. Will cross-feeding between glycolytic and OXPHOS cells alter the resulting spatial distribution of these cells? This could be addressed by adding a new type of nutrient produced by P_G that feeds P_O. The authors state in the Discussion that "we can also interpret these nutrients as mutually beneficial cell substrates produced by the glycolytic cells", but this should be directly evaluated. Nutrients produced by glycolytic cells are unlikely to contribute to proliferation of the glycolytic cells, but the model didn't distinguish between nutrients.*

The reviewer makes a good point about incorporating a metabolic symbiosis component to our models to determine what the predicted influences on patterning might be. We

have carried out these new modeling studies adding in a new term that models P_g cells feeding lactate to the P_o cells. Equation “3a” (Appendix A3.1 “Augmenting the Wnt Signaling Model”), incorporates lactate feeding to the P_o population and equation “3i” models the concentrations of lactate to depend on production by P_g and uptake by P_o . Appendix Table S5 (in Appendix A3.2) lists the corresponding parameters used in the modeling of Mock and dnLEF1 tumors. Appendix A3.2 shows the results of the model (Appendix Figure S12 and S13). Basically, we find that cross-feeding only weakly influences the dynamics of tumor growth and has little effect on the patterning. This is because the Wnt/Wnt inhibitor signaling dominates the patterning. We have also developed an in vitro version of the model (no blood vessels; Appendix A7) that incorporates the same type of cross-feeding and found similar results. We have reported these results in the main text (page 6, second paragraph starting with “We also considered a more general in vivo model...”), and in Figure 7D. Also, we provide further remarks in the Discussion (page 13, “When we considered the effects of symbiosis by explicitly incorporating cross-feeding between glycolytic and oxphos cells in a more general model ... we found that Wnt signaling dominates the behavior and the patterning is robust to this form of symbiosis.”)

2. The Wnt inhibitor/glycolysis inhibitor combination appears to be synergistic in a 3D in vitro system. However, the coexistence of glycolytic and OXPHOS cells in a regular spatial structure, as well as response to interference of Wnt signaling was validated by in vivo models. The 3D culture system may not be an appropriate system to test this model, since it lacks an important factor included in the mathematical model, that is, angiogenesis triggered by lactate produced in glycolytic cells, which increases nutrient availability of neighboring cells. I'm not sure if this factor is critical for the model to predict synergy between Wnt and glycolysis inhibition. It will be better if the authors could validate the synergy in in vivo tumor models or show that the synergy is robust to whether this term is included.

The reviewer highlights an aspect of our work that represents a major goal: our mathematical model predicts that specific combination therapies that target Wnt and metabolism will be synergistic in treating tumors *in vivo*. Within our current capabilities, we carried out 3D colony experiments to test this prediction and the results are encouraging. A major goal over the next two years is to fully test that prediction in mouse PDX models (subcutaneous and orthotopic) comparing responses between tumors that have a high glycolytic index and tumors that have a lower level of Wnt pathway activity and therefore, a lower glycolytic index. While this is an important goal, it is also challenging in time frame and cost to respond here, and we trust that the reviewers agree with us that it lies beyond the scope of this current study. To provide a more direct comparison with the in vitro system, we developed an in vitro version of the model where there was no angiogenesis but there was cross-feeding between the cell types (Appendix A7). We found that at low drug concentrations, there is significant synergy between XAV939 and DCA as measured by the Bliss Combination Index. The results are reported in the main text (page 11) and in Appendix A8.

Minor Comments

Reviewer 1:

- Figure 1A should demonstrate the core-periphery gradients, e.g. by adding a zoomed in version of the periphery as well or some quantification of this feature, which is currently missing.

We thank the reviewer for this suggestion as it has prompted additional analyses of the spotted pattern. We now show these analyses in Supplementary Figure S1. We include

images of the whole tumor, and we quantify the area of glycolytic spots from the periphery to the core of the tumor. In addition, we use Image J functions to show the change in spot density and overall pPDH positivity using a topographical map.

- Figure 1B is unclear (especially the zoomed in versions on the right). The image contains blue and red contours which are not explained (is this a result of automated detection)? At the minimum these contours should be added to the higher magnification panels.

We thank the reviewer for feedback on the best way to show both the native pattern and an overlay of our analysis. We have provided a more extensive description of the image analyses in the Appendix A1 (Supplementary Methods A1.1, A1.2), and the figure legend is modified to mention the image analysis with contour drawing with a mention and reference to supplemental methods. Figure 1A and 1B now show LEF1 and pPDH stains on serial sections, in part as a response to a major comment above, but also in part to respond to this issue. We prefer to keep the zoomed in version (40X) untouched by contour drawing so that readers can readily see the individual cells in the spots and the localization of the staining (cytoplasmic, peri-nuclear for phospho-PDH, nuclear (with some cytoplasmic) for LEF1). We have also changed the color of the boxes that indicate the enlarged portion of the panel. We hope readers can easily see the correspondence between the contours that our image analysis has drawn in the middle panel, and the localization of cell clusters with stronger staining.

In addition, we provide an expanded View Figure (EV1) of the same sections. In this figure the contours and convex hulls are shown without the IHC backdrop, and three different overlap calculations are shown. We trust that this further clarifies the arrangement of cell clusters and our image analyses.

- Figure 1D is also unclear, the authors should show a zoomed in version demonstrating areas of positive and negative staining. Some quantification is also needed to demonstrate that there is spatial heterogeneity in expression levels (how would staining look like in a regular tissue or a cell line). Without some positive or negative controls the panel shown does not demonstrate heterogeneity and it would actually be better to remove this entire claim.

Since we believe it is important to provide the reader with a comparison to what the heterogeneity of metabolism looks like primary human colon tumors, we have opted to respond to the Reviewer's request by providing additional images of normal tissue and primary tumor. Our new Figure 1D now includes an image of normal human colon crypt staining for pPDH and LEF-1. LEF1 is not expressed in normal colon epithelia and pPDH positivity is detected as a uniform gradient with the highest levels of detection in the crypt/stem cell region followed by decreasing positivity along the crypt into the differentiation zone. We have previously reported this pPDH staining pattern in Pate et al., 2014 EMBO 33(13):1454-73, but agree that it would be good to show additional images here. We have also juxtaposed new images of pPDH in colon tumors that better illustrate how pPDH positivity is heterogeneous. While quantification is difficult, the juxtaposed images of normal tissue and primary human colon cancer highlight that heterogeneity is more evident in tumor tissue. Although a spotted pattern is not as evident in the human colon tumor images, it is clear that there are clusters of cells that are positive for pPDH and they are directly juxtaposed to clusters of cells that are negative. We think that the modified figure better illustrates the idea of metabolic heterogeneity occurring in cell clusters and we thank the reviewer for suggesting that we make better attempts at showing this staining pattern. In addition, we now include serial sections of primary human colorectal cancer stained for pPDH and LEF-1 positivity. The staining patterns show remarkable concordance (Figure 1E).

- The text is missing a reference to Figure 1D when discussing the tumor staining.

Thank you – we have included the reference in the text.

- Figure 1 - the authors should quantify the size of the patches not only in microns but also in number of cells per patch.

We have quantified the number of cells in spots in Mock (7 cells) and dnLEF1 tumors (17 cells), and refer to this data in the text. We present the quantification in the Appendix A1 in a new Supplemental Figure S11 and referenced in the Results section when presenting figures 3D and 3E (page 8, second paragraph).

- Table of parameters in Figure 2 - The references are missing from the paper.

Thank you for catching this, we now include these references in the Bibliography.

- Model equations in Figure 2 - define N and the chi function.

Thank you for this suggestion, we have better described these functions in the main text (page 6, first paragraph). The precise forms of the chi functions are provided in Appendix A2, Supplementary Material, which we now refer to explicitly in the main text.

- Figure 4D,E - is there any change in the expression of glycolysis genes?

This was a great suggestion, and we carried out this analysis. We did indeed observe statistically significant increases in *ENO2* (2.7 fold), *PFKM* (43%) and interestingly *HIF1A* (1.98 fold; a transcription factor that also activates glycolytic metabolism). The lactate transporter *SLC16A3* and direct Wnt target gene *PDK1* were also increased (49% and 73% respectively), but with significant variation among the 13 patient datasets such that these increases did not reach statistical significance. This data is now presented as Expanded View Figure EV2.

- When presenting the mathematical model in the main text the authors should define W_i (the Wnt inhibitor) and S_w (the rate of Wnt production).

Thank you for catching this, we have modified the main text to define these terms in the discussion of Figure 2 (page 6, first paragraph).

- In the Discussion the authors state that the cells are genetically identical. Tumors cannot be assumed to be genetically identical due to clonal heterogeneity and indeed the authors later discuss alternative mechanisms, e.g. cell sorting that can be genetic in nature, the authors should explain this claim of non-genetic effects, e.g. by assuming a genetic bottleneck in the xenograft model.

The reviewer makes the good point that we cannot definitely rule out genetic heterogeneity due to the emergence of clonal populations. However, this possibility is extremely unlikely given that we consistently observe the spotted pattern, even in tumors developed for only 14 days, and when other cell lines (e.g. SW620) are used for xenografting. We have modified our text to say the following:

“Although we cannot rule out that the spotted pattern is due to the emergence of genetically distinct, clonal populations, the short timescale of the xenografting (14-21 days) and the reproducibility of the pattern in another cell line, as well as site of injection

(i.e. subcutaneous and orthotopic, Appendix A9, Supplementary. Figure S20), suggests that what we have observe is a fundamental pattern of tumor heterogeneity that is not genetic in nature, but non-genetic and dynamic.”

- *The authors should elaborate on the method of spot detection in the Methods section.*

Thank you for suggesting this. The methods section now includes a description of the algorithm for automatic spot detection, as well as the specific parameters used in the image processing.

- *The caption of Figure 3 contains several typos.*

Thank you for catching this – the typos have been corrected!

- *The choice of 161um for a scale bar in the images is quite unconventional.*

We have modified the scale bars to indicate 100um

-*Figure caption 6 mentioned B.1e and B.2e which I could not find.*

Thank you for catching that, we have fixed these typos.

Reviewer 2:

Given the importance of SW as a surrogate of Wnt signaling activity, it is important to explain this parameter earlier in the text (it is currently introduced on 7th page).

Thank you for noting this, we have now included a description while introducing the Reaction Diffusion model (Figure 2, page 6, first paragraph).

Figures 1-3: It would be useful to score the IHC staining just to show the level of heterogeneity in the patterning, especially post lentiviral transduction.

This was a great suggestion. To respond to this request, we developed a scoring matrix, and enlisted 3 unbiased observers to score three images each of Mock and dnLEF tumors stained for pPDH. We asked the observers to score the number of spots per image, the intensity of each spot, and the number of cells per spot. The data are shown in Appendix A1.15, Supplemental Figure S11. The number of cells per spot shows highly significant differences between Mock and dnLEF (7 cells vs. 16 cells), and the number of spots scored per image was also highly consistent,

The mean IHC intensity of pPDH in Mock and dnLEF1 spots is nearly the same, but a caution about direct comparison of intensities is important to mention because the IHC staining was performed independently for Mock and dnLEF1 tumors, and therefore color development of the stain was performed subjectively to minimize background. Even so, and interestingly, the distribution of intensities and cells per spot was greater in the dnLEF tumors, indicating that there may be an increase in the heterogeneity of the pattern.

Figure 3E: It would be more appropriate to average the data points for subsequent comparison with the simulation.

Thank you for catching this inadvertent omission. We have now added in the experimental averages onto the scatter plot in Figure 3E.

Figure 6C: What happens to combination response when the Wnt signaling is reduced by more than 30%? 50%? Etc. More rigorous metrics to assess synergy between the treatments would improve this analysis (e.g. the Bliss score or another one). Also, the right panel of Figure 6C is not explained clearly. What is the difference between $(S_W)_{\sim}$ and $(S_W)_{\square}$, which seems to be the only difference between the red dashed lines and the red solid lines?

We have included a new figure that shows a broader range of $\overline{S_W}$ (or, strength of Wnt signaling (Figure S18 in Appendix A8) as part of our effort to determine the effectiveness of combination therapy and synergy between the drugs. The modeling shows that XAV939 treatment alone is sufficient to kill the tumor only when Wnt signaling is reduced by 60% or more (e.g. $\overline{S_W} = 0.4$). Similarly, if DCA concentrations are high enough (eg. $1/\tau = 30$), this agent alone is enough to kill the tumor.

As for direct assessment of synergy, we are thankful to the Reviewer for suggesting use of the Bliss score and other measures. We calculated a measure of synergy using the Bliss Combination Index (BCI) (Foucquier and Guedj, 2015). As we now report in the main text, in the case of simulation of in vivo tumorigenesis and the experimental in vitro tumor spheroid growth in fibrin, the BCI is zero for all cases because individual treatments do not reduce tumor size, only the combination kills the tumor (Figure 6C and Figure 7C respectively). Any value less than 1.0 indicates some propensity for synergy with a value of zero indicating maximum synergy. In the case of simulation of the in vitro tumor spheroid growth, the Bliss score is 0.3462 (using $\overline{S_W} = 0.8$. $1/\tau_{go} = 1/4$; See Figure 7D and Appendices A7 and A8).

Finally, we have added a correction in the figure caption and main text to clarify the dashed and solid red lines in Figure 6C (right panel).

As for panel Figure 6C, The different between the two Wnt is ...??

We have added text to the legend and the results to better explain the difference between the two S_W terms: $(S_W)_{\sim}$ and $(S_W)_{\square}$. The tilde nomenclature depicts what happens to the effectiveness if we prevent “Wnt spreading” (ie. keep Wnt and Wnt inhibitor ligand diffusivities the same as they are in Mock).

It is suggested to include a figure showing the experimental results on combinational inhibition of Wnt and PDK1 in the manuscript instead of putting the related figures in Supplementary Materials which makes these results difficult to follow.

We have now created a new Figure 7 (old figure S28) showing these experimental results in the body of the manuscript. We have also added in vitro model results to this figure.

Appendix Figure S25 & S28: should clarify what statistical tests were used.

Thank you for pointing out this oversight. We have added a description of the statistics used for the analysis in the figure captions.

Thank you again for sending us your revised manuscript. We have now heard back from the two reviewers who agreed to evaluate your study. As you will see below, the reviewers think that most issues have been satisfactorily addressed. However, reviewer #1 lists two remaining concerns, which we would ask you to address in a minor revision.

REFEREE REPORTS

Reviewer #1:

The authors have significantly improved the paper and satisfactorily addressed most of my points. However, the major point of demonstrating the co-expression of glycolysis and Wnt is still lacking. The human tissue sections staining in Figure 1E are indeed striking, however the mock xenografts are not. In Figure 1B the authors have (perhaps by mistake) placed the automatic spots detected for the LEF1 staining on the phosphorylated PDH staining. This panel currently demonstrates little overlap between the spots. While these are serial sections and some of the 'overlapping' cells are not really the same cells, the authors currently cannot draw a firm conclusion of co-expression from these images. The quantification of overlap is also incorrect. Any two cellular features that stain 80% of the tissue would have 64% overlap even if there is no spatial correlation. The invariance to spot threshold selection is not convincing since the spot areas and numbers do not seem to substantially change in EV1. Also, the spots are quite large and as I understand from EV1 even if a small part of them overlap this is considered as a full overlap. In fact, if one would calculate the total blue area in EV1 and divide by the red or green areas the real overlap might even be lower than expected by chance.

A correct way to quantify overlap would be to identify the nuclei that appear in both serial sections, count what fraction of them have LEF1 staining in one, what fraction have phospho-PDF staining in the second section and count what fraction have both, then perform a hypergeometric test to obtain a p-value. I realize that this is a difficult system and identifying overlapping sufficient cells that appear in both sections may be hard. If the authors succeed to convince that there is a significant overlap this would substantially improve their paper, however even without this point I feel that the paper's theoretical approach is interesting and important as are some of the experimental validations and I support publication. However, if the authors fail the overlap validation they must tone down the discussion of this experimental finding in the text.

Minor comment: Figure 4D,E uses a parametric test, a nonparametric test should be performed instead of the t-test which assumes normality. Wilcoxon rank sum test could be appropriate here. In addition, multiple hypothesis correction must be applied (e.g. False Discovery Rate correction) instead of the 0.05 p-value threshold currently used.

Reviewer #2:

The authors have addressed my concerns in entirety. This will be an important paper.

Response to Reviews

Once again we thank the reviewers for their positive comments and valuable suggestions on improving the qualitative and quantitative aspects of this study. Below we provide some comment to the request for additional revisions and we describe the work we performed in response to these requests. We also submit a

revision in which new text is indicated in blue font. We trust that these revisions are acceptable to the reviewers and the editorial board.

Major Comments

1. Reviewer 1:

The authors have significantly improved the paper and satisfactorily addressed most of my points. However, the major point of demonstrating the co-expression of glycolysis and Wnt is still lacking. The human tissue sections staining in Figure 1E are indeed striking, however the mock xenografts are not. In Figure 1B the authors have (perhaps by mistake) placed the automatic spots detected for the LEF1 staining on the phosphorylated PDH staining. This panel currently demonstrates little overlap between the spots. While these are serial sections and some of the 'overlapping' cells are not really the same cells, the authors currently cannot draw a firm conclusion of co-expression from these images.

First of all, we wish to thank Reviewer #1 for catching the wrong image panel in Figure 1B. It is a testament to the careful, rigorous work of the Reviewer and we are deeply appreciative. The correct IHC image of LEF1 staining is now in place. The accidental overlay of LEF-1 contours on the pPDH IHC staining image highlights the reviewer's point that trying to align patterns from serially stained sections is challenging. Below we provide some counterpoint, but state here at the outset that we have taken Reviewer 1's advice and toned down the text – pointing out the caveats of serial sections and stating that we cannot definitely determine whether the heterogeneous patterns of pPDH and Wnt are one in the same.

The quantification of overlap is also incorrect. Any two cellular features that stain 80% of the tissue would have 64% overlap even if there is no spatial correlation. The invariance to spot threshold selection is not convincing since the spot areas and numbers do not seem to substantially change in EV1. Also, the spots are quite large and as I understand from EV1 even if a small part of them overlap this is considered as a full overlap. In fact, if one would calculate the total blue area in EV1 and divide by the red or green areas the real overlap might even be lower than expected by chance. A correct way to quantify overlap would be to identify the nuclei that appear in both serial sections, count what fraction of them have LEF1 staining in one, what fraction have phospho-PDF staining in the second section and count what fraction have both, then perform a hypergeometric test to obtain a p-value. I realize that this is a difficult system and identifying overlapping sufficient cells that appear in both sections may be hard. If the authors succeed to convince that there is a significant overlap this would substantially improve their paper, however even without this point I feel that the paper's theoretical approach is interesting and important as are some of the experimental validations and I support publication. However, if the authors fail the overlap validation they must tone down the discussion of this experimental finding in the text.

We understand the reviewer's points and incorporated additional tests to determine the statistical significance of an association between the pPDH and LEF-1 spots. Although we attempted to find direct matches between LEF-1 positive nuclei in one image with pPDH-positive cytoplasm in another image, we found it difficult to confidently assign a sufficient number of bisected cells in the serial images. Instead, we used our image analysis. Since the 4x images (leftmost) in Figs. 1A and 1B have the same size and the images were assumed to be lined up as close as possible, we can approximately pair the pixels in the images by assuming that each pixel location in one slice corresponds to the same pixel location in the other slice. In each image, a pixel inside a red contour was considered positively stained and a pixel outside a red contour was considered negatively stained. For each pair of pixels, we can construct a 2x2 contingency matrix that describes the staining, e.g., both pPDH and LEF-1 are positively stained, negatively stained or one is positive while the other is negative. Using the Cochran-Mantel-Haenszel test, we can conclude that the pPDH and LEF-1 spots are significantly associated with each other with $p < 0.0001$. We have updated the main text and the supplementary material (A1.2) to describe this analysis. Because this analysis does guarantee that the paired pixels are in the same cell and also does not take into account differences

in spot densities in space, we have also toned down our discussion on the data shown in Figures 1 and EV1 as the reviewer suggests.

Minor Comments

Minor comment: Figure 4D,E uses a parametric test, a nonparametric test should be performed instead of the t-test which assumes normality. Wilcoxon rank sum test could be appropriate here. In addition, multiple hypothesis correction must be applied (e.g. False Discovery Rate correction) instead of the the 0.05 pvalue threshold currently used.

We thank the Reviewer for directing us to a different statistical test for significance in Figure 4D,E. In this revision we have used the nonparametric Mann-Whitney U test and have coupled it with a Benjamini-Hochberg correction for multiple hypothesis testing (RStudio). We have also re-analysed the data shown in expanded view Figure EV2. The outcome of this type of analysis is that the significance for a couple of the genes falls away. Interestingly however, the changes in the expression of Wnt-diffusion regulators remains quite significant. In total, four Wnt-diffusing regulators (*GPC1*, *SFRP1*, *SFRP2* and *SFRP4*) increase in expression *specifically* in radiochemotherapy-treated tumor tissue – not in the treated normal tissue. In Figure EV2, none of the increases in Wnt ligand expression pass significance if $p < 0.05$ is used as a cut-off, however we note in the legend that there is a trend for *WNT5B*, *WNT8B* and *WNT10B* to be increased with p-values < 0.10 . We think this is worth noting and we give the specific values in the Figure legend to point out where adjusted p-values fall in relation to 0.05 and 0.10. Similarly, one of the glycolysis genes, *ENO2*, is significantly increased in tumor tissue (p-value = 0.008) while the increase in the glycolysis regulator *HIF1A* is less so (p-value = 0.06). We also provide these numbers in the legend because we feel that this gives the reader the maximum amount of information so that he/she can judge for themselves whether or not the changes that occur in this dataset are notable. The new statistical test is more rigorous, and it therefore more strongly authenticates the major conclusion of the study which is that regulators of Wnt ligand diffusion are significantly increased in tumors subjected to radiochemotherapy – and therefore, the dynamics of Wnt signaling and metabolism are definitely changing – most likely as a survival response to the stresses of the therapy. Once again, we thank the Reviewer for asking for this analysis as it has strengthened the manuscript.

3rd Editorial Decision

12 January 2017

Thank you for sending us your revised manuscript. We are now satisfied with the modifications made and I am pleased to inform you that your paper has been accepted for publication.

Corresponding Author Name: John Lowengrub
 Journal Submitted to: Molecular Systems Biology
 Manuscript Number: MSB-2016-7386